# Structural insights into opposing actions of neurosteroids on GABA<sub>A</sub> receptors

Dagimhiwat H. Legesse[1], Chen Fan [2], Jinfeng Teng[3], Yuxuan Zhuang [2], Rebecca J. Howard [2], Colleen M. Noviello[3], Erik Lindahl [2,4] ✉ & Ryan E. Hibbs [1,3] ✉

γ-Aminobutyric acid type A (GABA<sub>A</sub>) receptors mediate fast inhibitory signaling in the brain and are targets of numerous drugs and endogenous neurosteroids. A subset of neurosteroids are GABA<sub>A</sub> receptor positive allosteric modulators; one of these, allopregnanolone, is the only drug approved specifically for treating postpartum depression. There is a consensus emerging from structural, physiological and photolabeling studies as to where positive modulators bind, but how they potentiate GABA activation remains unclear. Other neurosteroids are negative modulators of GABA<sub>A</sub> receptors, but their binding sites remain debated. Here we present structures of a synaptic GABA<sub>A</sub> receptor bound to allopregnanolone and two inhibitory sulfated neurosteroids. Allopregnanolone binds at the receptor-bilayer interface, in the consensus potentiator site. In contrast, inhibitory neurosteroids bind in the pore. MD simulations and electrophysiology support a mechanism by which allopregnanolone potentiates channel activity and suggest the dominant mechanism for sulfated neurosteroid inhibition is through pore block.

Neurosteroids regulate neuronal activity in the central and peripheral nervous systems. The levels of endogenous neurosteroids in the body are dynamic and altered by stress, pregnancy, the ovarian cycle, neural development, and aging[1]. Their dysregulation can result in neurological and psychiatric disorders; accordingly, neurosteroids are employed clinically as sedatives, hypnotics, anticonvulsants, and antidepressants[2–4]. These steroids modulate neurotransmission in the brain by interacting with a variety of membrane proteins including ionotropic γ-aminobutyric acid type A (GABA<sub>A</sub>) receptors. One such neurosteroid, allopregnanolone, is the first FDA-approved drug to treat post-partum depression.

GABA<sub>A</sub> receptors belong to the Cys-loop superfamily of pentameric ligand-gated ion channels. GABA<sub>A</sub> receptors mainly assemble as heteromers, and each subunit shares a common topology, with a large N-terminal extracellular domain (ECD), followed by four transmembrane α-helices (M1-M4; TMD). A poorly conserved intracellular domain (ICD) connects the M3 and M4 helices. GABA binds at β-α

subunit interfaces in the ECD[5,6]. The ICD is disordered in experimental structures and includes motifs important for intracellular sorting and plasma membrane localization. Binding of GABA to a resting-state receptor results in conformational changes that allow anions, mainly chloride, to pass through the intrinsic TMD channel and, in most cases, inhibit neuronal excitability[7]. In the sustained presence of GABA, the receptor then enters a desensitized state wherein agonist remains bound, and the channel adopts a closed, non-conducting conformation distinct from that in the resting state. The TMD is surrounded by lipids, and allosteric modulation of GABA<sub>A</sub> receptors by several drug classes occurs in the TMD[8–10]. These allosteric modulators include important exogenous molecules, like general anesthetics and sedative-hypnotics, and diverse endogenous molecules, like neurosteroids.

Our current understanding of GABA<sub>A</sub> receptor modulation by neurosteroids comes from photoaffinity and binding experiments, crystallographic structures, and electrophysiology. These studies have shed light on the subunit dependence of neurosteroid modulation and

[1]Department of Neuroscience, UT Southwestern Medical Center, Dallas, TX, USA. [2]Dept. of Biochemistry and Biophysics, Science for Life Laboratory, Stockholm University, Solna, Sweden. [3]Department of Neurobiology, University of California San Diego, La Jolla, CA, USA. [4]Dept. of Applied Physics, Science for Life Laboratory, KTH Royal Institute of Technology, Solna, Sweden. ✉e-mail: erik.lindahl@scilifelab.se; rehibbs@ucsd.edu

amino acid determinants of potentiation and inhibition[11-21]. While crystallographic structures and functional measurements paint a consistent picture of where positive allosteric modulator (PAM) neurosteroids bind, there is a lack of consensus on the mechanism for inhibitory neurosteroids (or negative allosteric modulators, NAM). Where NAM neurosteroids bind, and how these often-sulfated molecules access suggested hydrophobic membrane sites, remain poorly understood. Here, we combine cryo-electron microscopy (cryo-EM) studies with electrophysiology and molecular dynamics (MD) simulations to study GABA_A receptor binding sites and allosteric mechanisms for PAM neurosteroids versus sulfated NAM neurosteroids. Our findings reinforce earlier conclusions about PAM neurosteroids, revealing a TMD subunit interface site for allopregnanolone. Structural analysis and MD simulations suggest how binding of this drug increases sensitivity to GABA and increases ion channel width, thereby enhancing receptor activity. Investigations of two NAM neurosteroids reveal the principal site of inhibition to be in the ion channel, rather than at previously suggested sites at the receptor-lipid interface. We reconcile this finding with earlier structural and functional data to propose an intuitively simple mechanism of antagonism by sulfated NAM neurosteroids.

## Results

### Structure determination of receptor-neurosteroid complexes

Neurosteroids are the most potent endogenous modulators of GABA_A receptors[22]. Endogenous neurosteroids are synthesized from cholesterol in neurons and glia, where a series of enzymatic reactions add or remove functional groups to the steroid backbone that result in derivatives with a spectrum of activities[23]. We used cryo-EM and electrophysiology to better understand GABA_A receptor modulation and binding interactions of three particularly important neurosteroids: the PAM allopregnanolone, and two NAMs, pregnenolone sulfate (PS), and dehydroepiandrosterone sulfate (DHEAS) (Fig. 1). We followed a similar approach to determine structures of all three complexes (Methods, Supplementary Figs. 1–2). We first purified a modified α1β2γ2 GABA_A receptor construct, in which the intracellular loops were removed, and reconstituted it into lipid nanodiscs[24]. We then formed a complex with GABA, neurosteroid, and Fab fragments to aid in particle alignment, and collected single particle cryo-EM datasets.

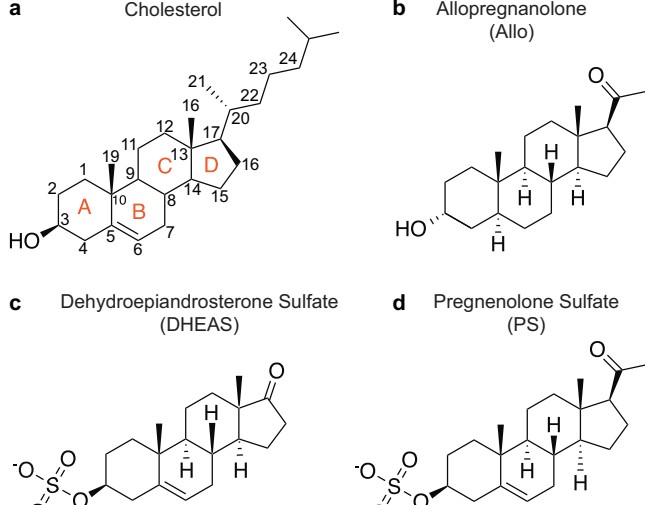

**Fig. 1 | Neurosteroid chemical structures. a** The neurosteroid precursor cholesterol with rings and atom numbers indicated. **b–d** Allopregnanolone is a GABA_A receptor positive modulator while the sulfated neurosteroids, DHEAS and PS, are negative modulators.

### Allopregnanolone binds at β-α interfaces in the TMD

We selected allopregnanolone among the PAM neurosteroids due to its remarkable clinical application in depression and an absence of structural information for its binding to GABA_A receptors. Note, while preparing this manuscript, a preprint was made available on allopregnanolone bound to native GABA_A receptors[25]. Allopregnanolone acts as a positive allosteric modulator of GABA_A receptors. During pregnancy, the levels of hormones like progesterone, a precursor for allopregnanolone, increase over 100-fold[26]. To avoid sedation, the expression level of GABA_A receptors drops[26]. After birth, hormones rapidly return to pre-pregnancy levels; however, the level of GABA_A receptors does not recover at the same rate[27]. This lag in changing expression levels compounded with fluctuations in the level of allopregnanolone in the brain is associated with mood disorders, including premenstrual dysphoric disorder and postpartum depression[28]. To probe allopregnanolone's mechanism of action, we determined the structure of the α1β2γ2 GABA_A receptor in complex with GABA and allopregnanolone at 2.9 Å resolution (Supplementary Figs. 1, 2a–c). The map quality was sufficient to model all the expressed receptor construct and position allopregnanolone in its TMD sites (Fig. 2a). Importantly, we confirmed neurosteroid potentiating activity in the cryo-EM construct (Fig. 2b).

We were interested to study the allopregnanolone mechanism at two levels: first, where does the drug bind, which residues does it contact, and are these interactions important for drug activity? Second, how does allopregnanolone binding influence the receptor structure and dynamics, locally and at a distance from the binding site, and how do these effects contribute to potentiation? To answer the first question, we examined the atomic interactions of allopregnanolone. We observed sausage-like densities in a hydrophobic cavity at the two β2-α1 subunit interfaces, near the base of the TMD (Fig. 2c). These densities correspond to an established binding site for potentiating neurosteroids and are not found in the absence of allopregnanolone[11,13,29]. The nature of these sites, where more than half the surface area of interaction is made by bulk lipids, results in ligand density lacking strong high-resolution features. Accordingly, we used MD simulations to test the stability of different allopregnanolone poses that fitted the density comparably well (Supplementary Fig. 3). One orientation (pose 1) was far more stable than all others, in which the steroid's A ring tilts toward the M1 helix of the α subunit and the D ring points toward the cytosolic end of M3 of the β subunit (Fig. 2c). The C3 hydroxyl group is positioned to form a hydrogen bond with the epsilon oxygen of αQ242 (Fig. 2c). This interaction is essential, as the αQ242L mutation ablates the potentiating effects of neurosteroids, but other mutations that preserve the H-bond do not alter PAM neurosteroid activity[30–32]. This well-defined interaction, along with the experimental density and MD results, allowed us to confidently orient the D ring C20 ketone toward L301 of the adjacent β2 subunit. This hydrophobic residue is conserved among all GABA_A receptor α subunits. The indole side chain of W246 is positioned to make stacking interactions with the C and D rings of allopregnanolone, stabilizing the steroid in the binding pocket. The importance of this interaction is underscored by its mutation to leucine resulting in complete loss of potentiation by allopregnanolone and pregnanolone[20,33,34] with no effect on barbiturate potentiation[30].

### Impact of allopregnanolone binding on receptor conformation and GABA stability

In cell membranes, binding of allopregnanolone to the GABA_A receptor enhances channel activation by GABA. We compared the α1β2γ2 GABA_A receptor structure bound to GABA alone (PDB: 6X3Z) versus that bound to GABA and allopregnanolone to identify differences that may contribute to potentiation (Fig. 3a). Global superposition revealed small differences, with a Cα root mean-squared deviation (rmsd) at 0.66 Å between the two structures. There are notable differences in the

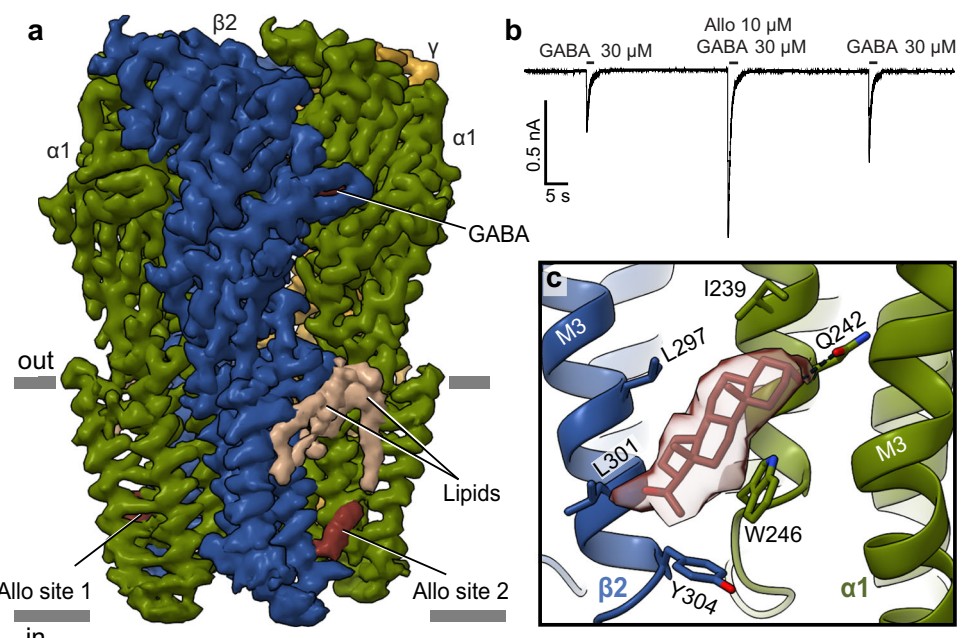

**Fig. 2 | Allopregnanolone activity and binding site. a** Density map of the GABA_A receptor bound to GABA and allopregnanolone with ligands and lipids indicated. Subunits are colored in dark green (α1), blue (β2), and gold (γ2), with lipids in beige. **b** Whole cell patch-clamp electrophysiology showing allopregnanolone potentiation of GABA response in the cryo-EM construct, similar responses were seen in $n = 8$ independent cells. **c** Detail for allopregnanolone binding at one β-α interface. Subunits are colored as in (**a**). Ligand is shown in red sticks with experimental density shown as a semi-transparent surface.

alignment of the transmembrane helices (Fig. 3b) that stem from local changes in conformation surrounding the allopregnanolone site. In the ligand-bound conformation, the stacking W246 shifts intracellularly, accommodating the bulky steroid (Fig. 3c). The M1M2 loop, just below this M1 tryptophan, pivots away from the pore axis, coincident with altered interactions between αM1M2 and βM3. These conformational transitions near the neurosteroid site result in loosening of TMD subunit interfaces that enable tilting and twisting of several transmembrane helices in cryo-EM structures in the presence versus absence of allopregnanolone (Fig. 3b). The outward-facing end of the pore is stabilized through new electrostatic interactions. Specifically, in the GABA bound structure, E270 of βM2 orients away from the pore and interacts with the N275 of the adjacent α1 subunit (Fig. 3d). In the presence of allopregnanolone, E270 points toward the pore and interacts with βH267. Together, the M2 helix movement results in apical pore dilation observed in the presence of allopregnanolone (Fig. 3e, f).

Comparisons of pore conformation lend insight into overall mechanisms of positive modulation. In the GABA-alone structure, the primary pore constriction is at the −2′ position, a structural hallmark of a non-conducting desensitized state (Fig. 3e). There is a secondary constriction at the activation gate found at the 9′ position of the pore with a diameter of 4.6 Å. Examination of the pore diameters in the GABA + allopregnanolone model indicate conservation of the −2′ constriction but an expansion of the 9′ constriction to approximately 9.1 Å (Fig. 3f). In simulations, this expansion was associated with increased hydration of the pore (Fig. 3i), an effect consistent with increased ion access to the hydrophobic constriction. Indeed, the free-energy barrier for chloride permeation across the 9′ gate was ≤18 kJ/mol in the presence of allopregnanolone (Fig. 3j), substantially lower than in the structure with GABA alone (≥32 kJ/mol)[24].

This effect is reminiscent of other PAM-bound complexes, including etomidate and propofol, which widen the pore diameter at the 9′ position to 8.4 Å and 10.5 Å, respectively (Fig. 3g, h)[24]. Indeed, the energy barrier at 9′ is lowered nearly as far with allopregnanolone as was previously shown with propofol (≤7 kJ/mol) (Fig. 3j)[24]. The widest

diameter of the ion pore is found near its extracellular end, between 17′ and 20′. This more open state may be a common feature of positive allosteric modulators, as it is seen across the Cys-loop receptor superfamily[10,35]. A caveat of these interpretations is that all PAM complexes studied to date adopt non-conducting, desensitized-like conformations.

The presence of allopregnanolone at the cytosolic end of the TMD appears to have long-range effects on receptor dynamics, including in the ECD almost 80 Å away. Simulations in the presence of allopregnanolone revealed reduced flexibility and overall spread of the outer ECD (Fig. 3k), phenomena associated with activation in pentameric ligand-gated ion channels[35,36]. Interestingly, GABA dissociated from its extracellular binding site in multiple simulations of the GABA-only structure, but not in any of our simulations in the presence of allopregnanolone (Fig. 3l, Supplementary Fig. 4), reminiscent of the agonist-stabilizing effect of diazepam in previous independent simulations[24]. In summary, the presence of allopregnanolone in its binding pocket at the intracellular end of the transmembrane domain influences the conformational state and dynamics of the entire receptor including changes from the pore to the outer ECD consistent with increased activity.

### Structural basis of sulfated neurosteroid inhibition

Sulfated neurosteroids are negative allosteric modulators of GABA_A receptors[37,38]. The best-studied endogenous sulfated neurosteroids are pregnenolone sulfate (PS) and dehydroepiandrosterone sulfate (DHEAS), both of which play important roles in memory, learning, and aging[37]. DHEAS and PS share similar chemical structures and differ only in the C17 carbon substituent of the sterol D ring (Fig. 1c, d): the carbonyl oxygen of DHEAS is substituted with an acetyl group in PS. These steroids also differ in their negative modulatory effects on GABA_A receptors. Overall, they both inhibit GABA_A receptor activity[38]. However, DHEAS exerts its effects more by decreasing the current amplitude with little effect on kinetics, while PS both reduces peak currents and accelerates apparent desensitization, in both the full length and cryo-EM constructs (Fig. 4a, b)[37]. The binding site(s) for sulfated

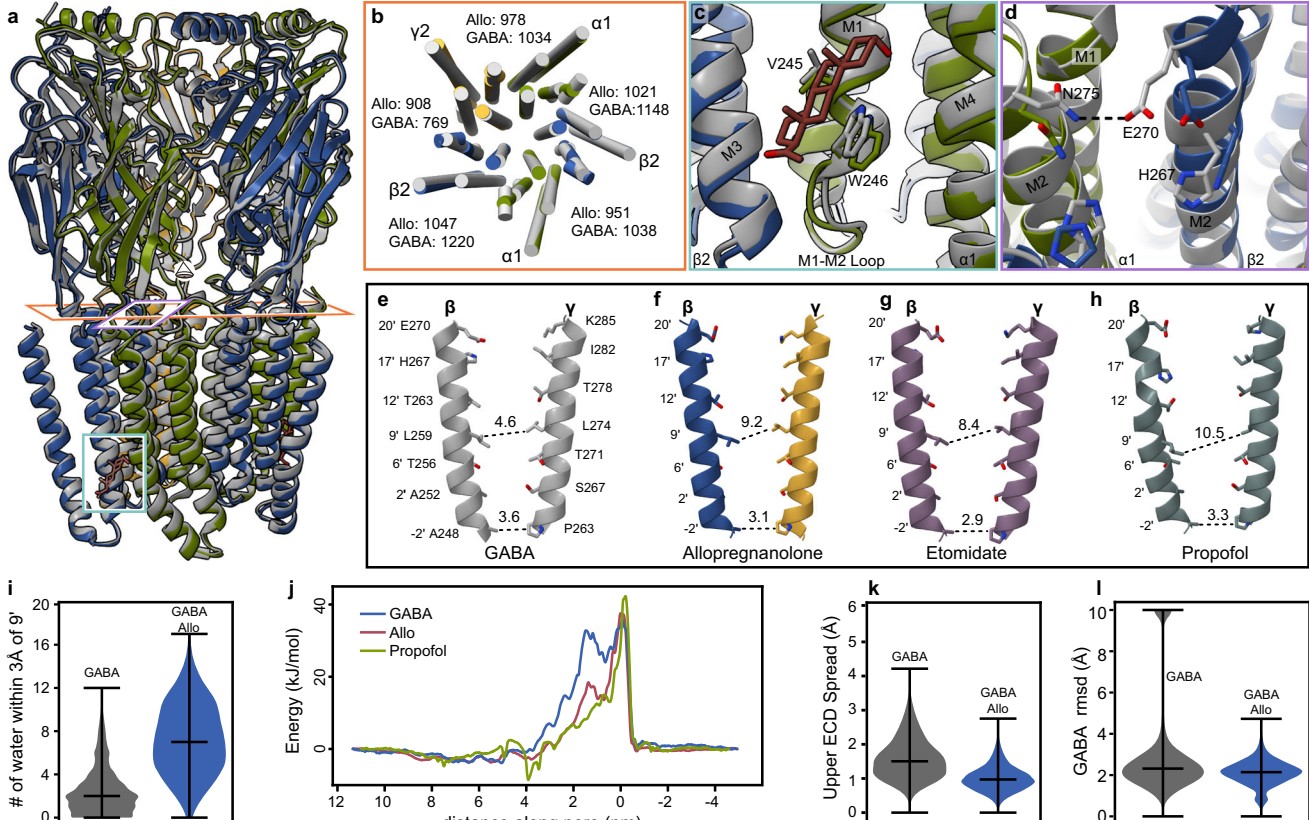

**Fig. 3 | Allopregnanolone potentiation mechanism. a** Superposition of allo-pregnanolone (Allo) complex, colored as in Fig. 2, with published structure of the receptor bound to only GABA (gray, 6X3Z). Boxes indicate areas highlighted in (**b**–**d**). **b** Horizontal slice through TMD shown from extracellular perspective, with subunits colored as in (**a**). Numbers between subunits indicate surface area buried (Å²) measured by PDBePISA. **c** Allopregnanolone binding site detail, colored as in Fig. 1, compared to GABA-only structure (in gray). **d** Changes in rotamers at the top of the M2 helix lining the pore, colored as in (**c**). **e**–**h** M2 helices of β and γ subunits, with helices colored by structure: GABA complex in grey, allopregnanolone colored as in (**a**), etomidate in mauve, and propofol in dark grey. Numbers indicate pore diameters in Å at different positions measured by Hole[78]. PDB codes for each group: **e** 6X3Z; **g** 6X3V; **h** 6X3T. **i** Number of waters within 3 Å of the M2 9′ gate. The pore of the GABA+Allo system (blue) is more hydrated than the system with GABA alone (gray). **j** Free energies for chloride ion permeation along the pore axis (left–right

extracellular–cytoplasmic sides, with −2′ gate at 0 nm) for representative α1β2γ2 complexes. Overlaid plots show the energy barrier at the 9′ hydrophobic gate (around 1.5 nm) in the structure with GABA alone (blue) to be partially relieved in the GABA+Allo complex (red), nearly to the extent previously reported for GABA + propofol (green). **k** Upper-ECD spread during simulations in the presence of GABA (gray) or GABA+Allo (blue). Allo binding is associated with contraction of the upper ECD. **l** GABA stability as measured by rmsd during simulations in the absence (gray) or presence of Allo (blue). Upon exiting the orthosteric binding pocket, free diffusion of GABA was associated with extreme rmsd values (>60 Å), which were arbitrarily cropped to 10 Å for ease of comparison. No unbinding events were observed in the presence of Allo. Violin plots (**i**, **k**, **l**) represent probability densities from 4 independent simulation replicates of >400 ns each, sampled every 0.4 ns (*n* > 4000), with markers indicating median and extrema.

neurosteroids on GABA$_A$ receptors is controversial with conflicting findings over the past several decades from electrophysiology and structural biology studies[30,39–41]. We sought to resolve this controversy by determining structures of the α1β2γ2 GABA$_A$ receptor in complex with GABA + PS and GABA + DHEAS at 2.6 Å and 2.7 Å resolution respectively (Supplementary Fig. 2a–c).

To our surprise, both structures bound to NAM neurosteroids revealed multiple potential binding sites for these ligands (Supplementary Fig. 5a). First, we observed two strong densities in the TMD at β2-α1 interfaces that overlap with the allopregnanolone binding sites. Second, density was present in the benzodiazepine binding pocket of the α-γ interface in the ECD. Finally, there was a strong density in the ion pore for both NAM-bound datasets. To determine which site(s) are physiologically relevant, we examined them in the context of the existing literature, and electrophysiology and MD studies guided by our structural findings. We propose that the pore site is the most important inhibitory site of action for NAM neuro-steroids. We did not find evidence for an α-subunit TMD site postu-lated for pregnenolone sulfate[19] and instead find that this site is occupied by lipids.

## Interrogation of the TMD and ECD subunit interface sites for sulfated neurosteroids

When the receptor was prepared with either DHEAS or PS, we observed densities in the transmembrane domain that overlap with the allo-pregnanolone site. We attempted to build neurosteroids in this pocket between the M1 helix of α1 and the M3 helix of β2, with the D ring of DHEAS and PS facing the α1 subunit and the A ring toward the cytosolic TMD junction (Supplementary Fig. 5a–b). In this pose, the C17 carbonyl oxygen of DHEAS and the C20 acetyl oxygen of PS could form hydrogen bonds with the amide side chain of αQ242 (Supplementary Fig. 5c, d). The hydrophobic tetracyclic core of both ligands would be positioned to stack against αW246, similar to allopregnanolone, while the sulfate group could interact with L301 of the β2 subunit. To test the hypothesis that αQ242, αW246, βL301 are important for NAM function in cells, we mutated these positions and measured inhibition of GABA evoked currents (Supplementary Fig. 5e). None of the mutations, including the triple mutant, affected NAM-mediated inhibition. We next performed MD simulations to assess the stability of NAM neuro-steroids in the TMD β2-α1 subunit interface site. MD results support a lack of stability of sulfated neurosteroids in this site; they quickly drift

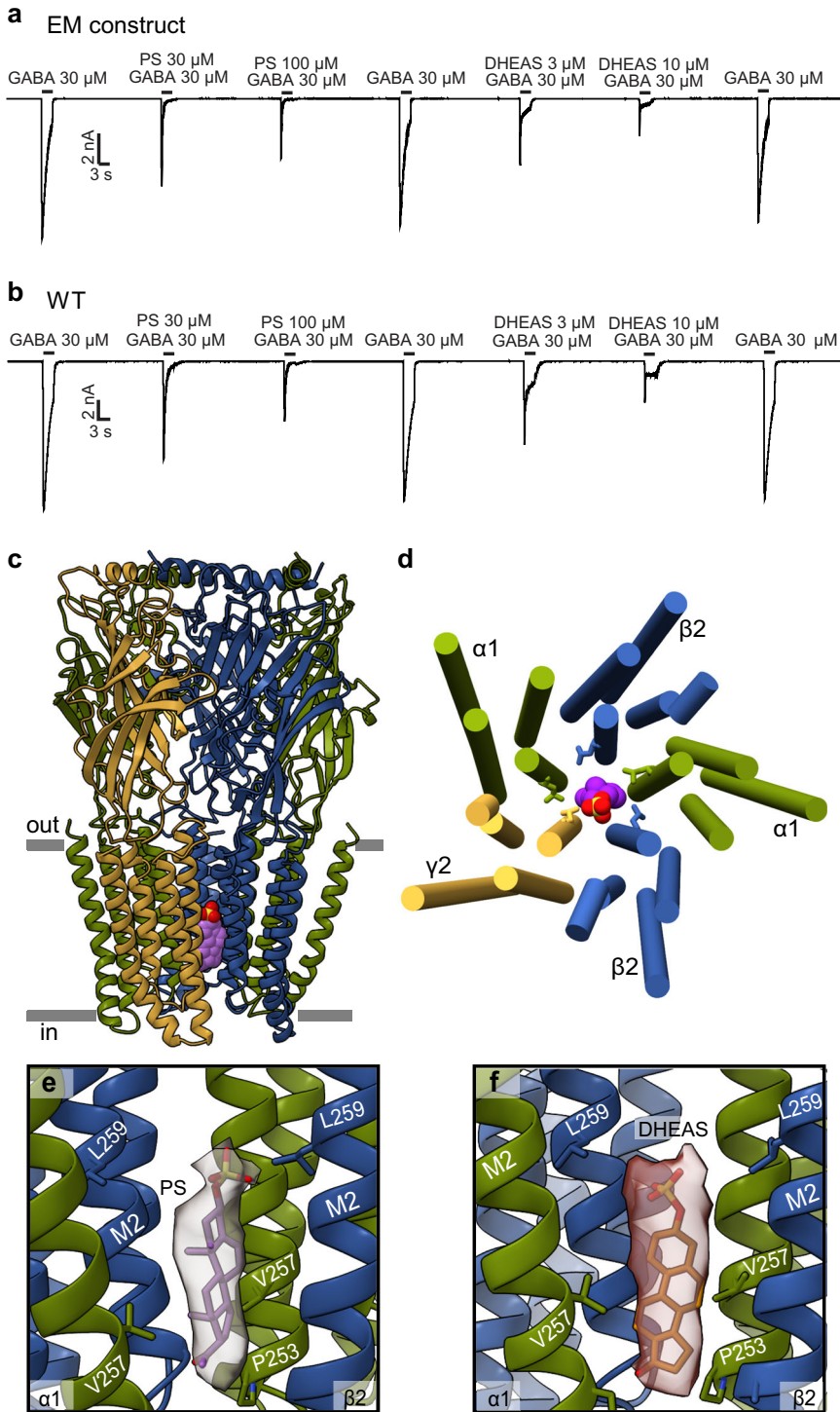

**Fig. 4 | Sulfated neurosteroids act as pore blockers. a, b** Whole cell patch-clamp electrophysiology recordings comparing the sulfated neurosteroid responses of the EM construct to the full-length WT receptor. **c** Side view of GABA$_A$ receptor – PS complex structure. **d** View of TMD from extracellular side with PS (colored spheres) bound in the pore. Side chains are shown for 9′Leu residues that form activation gate. **e** Detailed view of PS in pore site with experimental density shown as semi-transparent surface. **f** DHEAS in the pore as for PS in (**e**). In (**e**) and (**f**), the γ-subunit is removed for clarity. **c–f** Subunits are colored as in Fig. 2, with PS in purple and DHEAS in scarlet.

toward the intracellular side of the model membrane to interact with polar lipid head groups (Supplementary Fig. 5e–h). We suggest that the observed density at the PAM site in the PS and DHEAS-bound structures corresponds to a sticky steroid binding site that is not functionally coupled to channel activity and may arise as a function of the receptor preparation being in a lipid nanodisc, and not a planar bilayer.

The extracellular benzodiazepine binding pocket is the high affinity site for positive allosteric modulators such as diazepam, zolpidem, and alprazolam[24,42,43]. Occupation of this pocket by sulfated neurosteroids was unexpected, and thus we performed similar mutational studies and electrophysiology to examine the contribution of this site to inhibition by sulfated neurosteroids. We also assessed whether inhibition by PS and DHEAS could be blocked by flumazenil, which

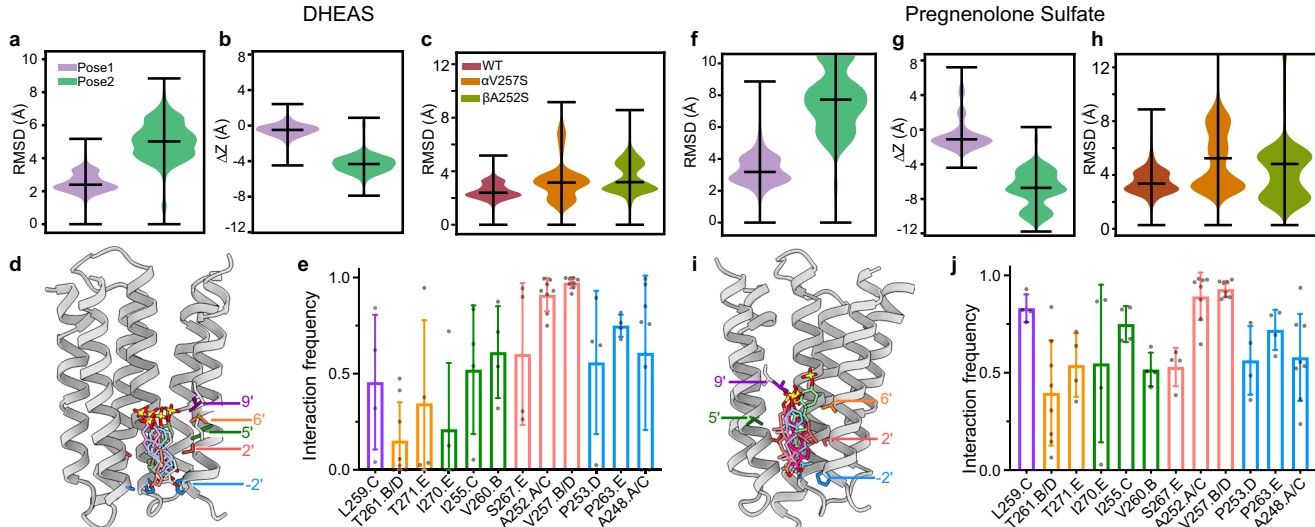

**Fig. 5 | Stability and interactions of sulfated neurosteroids in the pore. a** DHEAS stability based on rmsd in simulations initiated with the sulfate oriented up (purple) or down (green). **b** DHEAS movement along the channel pore (z-axis) during simulations. Positive and negative numbers indicate movement toward the outer and inner membrane leaflets, respectively. **c** DHEAS stability in simulations of WT, αV257S, and βA252S systems in the sulfate-up pose. Both mutations were associated with increased deviations relative to the starting pose. Violin plots (**a**–**c**) represent probability densities from 4 independent simulation replicates of >400 ns each, sampled every 0.4 ns ($n > 4000$), with markers indicating median and extrema. **d**, Representative DHEAS poses during simulations. Five poses were chosen by cluster analysis based on neurosteroid rmsd. Representative -2′, 2′, 5′, 6′, and 9′ residues along the righthand M2 helix are labeled and colored for reference to the bar graph in (**e**). **e** Interaction frequencies of residues in contact with DHEAS. Residues are grouped and colored according to panel (**d**). Each column indicates mean ± standard deviation of 4 independent simulation replicates ($n = 4$), with values from individual replicates as gray dots. **f**–**j** Analyses for PS as in panels (**a**–**e**).

occupies the benzodiazepine binding pocket and reverses the effects of diazepam and other benzodiazepines. Neither mutations in the α-γ ECD interface, nor competition with flumazenil, affected inhibition by PS and DHEAS (Supplementary Fig. 6). We conclude that this site is also irrelevant to the activity of sulfated neurosteroids.

## Sulfated neurosteroids bind in the pore

While allopregnanolone can diffuse through the membrane to its binding site in the TMD, the charged sulfate groups in PS and DHEAS hinder this membrane partitioning. PS is unable to inhibit GABA$_A$ receptors from the cytosolic side of the cell membrane, suggesting that the sulfated neurosteroids can only access their binding site(s) externally[44,45]. This finding, combined with our mutational studies, led us to examine the receptor cryo-EM maps for densities in additional places that may correlate to the physiologically relevant sulfated neurosteroid binding site. We observed a strong density in the ion pore of the PS + GABA and DHEAS + GABA data sets that is absent without the addition of sulfated neurosteroids to the preparations (Fig. 4c, d)[19–21,24]. These sausage-shaped densities follow the central axis of the pore from the −2′ position to the 9′ position, and exhibit features suggestive of the sulfate group orienting up, extracellularly (Fig. 4e, f). To test the stability and orientation preferences of sulfated neurosteroids in the channel, we first modeled two poses for each molecule, first with the A ring sulfate oriented toward the extracellular side and the D ring toward the base of the ion channel pore, and second, with the ligand flipped on its long axis, with the sulfate oriented intracellularly. MD simulations showed that the former orientation, with the sulfate up, is relatively stable, in contrast with a tendency for the neurosteroid to move toward the intracellular side when the sulfate is oriented down (Fig. 5a, b, f, g). This sulfate-up pose was preferred for both PS and DHEAS, despite subtle differences in the initial configurations of the two ligands.

Within the more stable sulfate-up simulations, neurosteroids, particularly PS, nonetheless sampled a range of orientations. After clustering individual simulation frames based on the relative rmsd of the neurosteroids, representative poses from the five most frequent clusters included various rotations along the steroid long axis (Fig. 5d).

Accordingly, both ligands interacted frequently (≥50%) with residues at the −2′ and 2′ positions in all α, β, and γ subunits (Fig. 5e, j). The most consistent contacts were with the 2′ residues in α (V257) and β (A252), where mutations have been shown to reduce NAM activity[40]. Indeed, the pore poses for both PS and DHEAS were destabilized in simulations with serine introduced at either αV257 or βA252 (Fig. 5c, h). Interestingly, whereas DHEAS interactions were primarily with these residues in the inner pore, PS contacts were distributed more broadly among positions −2′, 2′, 5′, 6′, and 9′ (Fig. 5e, j), and sampled greater deviations from the initial pose (Fig. 5d, i). Compared to DHEAS, a less fixed position of PS could reflect its lower apparent affinity for NAM activity at GABA$_A$ receptors[37]. Interestingly, we found that binding of sulfated neurosteroids in the pore stabilized a channel conformation distinct from that stabilized by GABA alone (Supplementary Fig. 7), consistent with a recent functional study concluding that PS and DHEAS stabilize an atypical nonconducting state[38]. We place this proposed pore block mechanism for sulfated neurosteroids in the context of earlier work in the Discussion.

## Neurosteroids influence receptor-lipid interactions

Interactions of neurosteroids like allopregnanolone with membrane lipids are important not only for accessing their binding sites but also for their stability and activity once bound[46,47]. The cryo-EM structures revealed several well-ordered lipid densities positioned to form a component of the neurosteroid binding sites, as well as lipid densities distal from the PAM neurosteroid site likely important for receptor stability (Figs. 2, 3, and 6). The pattern of bound lipids is different from that observed in previous GABA$_A$ receptor structures bound to endogenous neurotransmitter alone or other allosteric modulators.

We focus on four distinct lipid interactions in our best resolved PS-bound structure. These lipids can be categorized as outer-leaflet lipids located just below the ECD-TMD interface and inner-leaflet lipids situated at the base of the TMD (Figs. 2a, 6). The lipid occupancy for the first outer leaflet site present at both β-α interfaces was modeled as phosphatidylethanolamine (POPE), which is the most abundant lipid in our preparation (Fig. 6c). In the absence of endogenous neurosteroids, this lipid extends from the β2 M2-M3 loop deeply into the bilayer, with

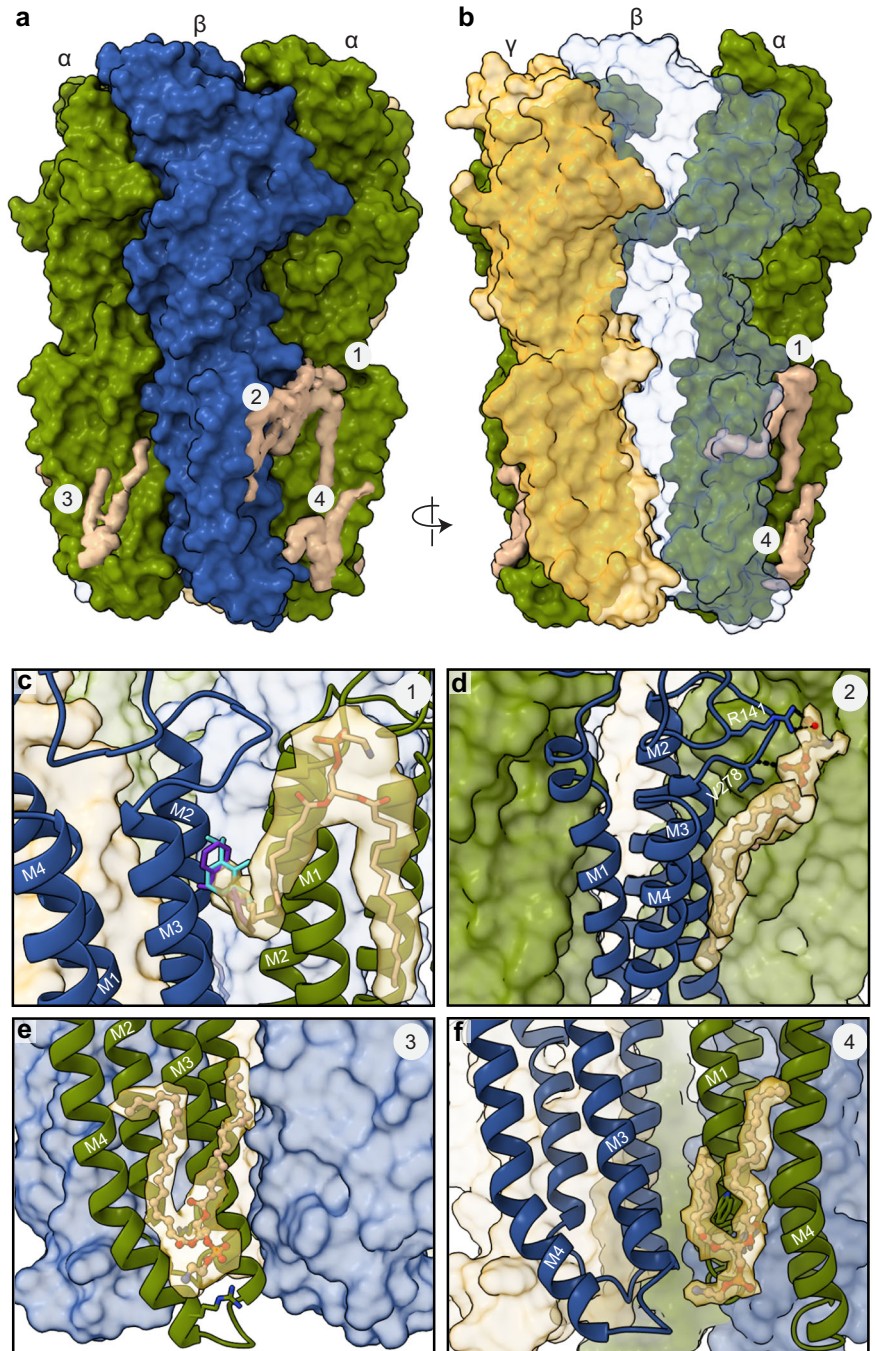

**Fig. 6 | Lipid interactions in GABA_A receptor – pregnenolone sulfate complex.**
**a** Side view of the GABA_A receptor in complex with PS. **b** Model rotated to highlight site 1 lipid protruding into β-α subunit interface. β2 subunit in translucent color. **c** Site 1 lipid tail protruding into where etomidate and propofol in purple and light blue were previously found to bind. **d** Site 2 lipid at the β2-α1 subunit interface. **e** Site 3 lipid binds in an α-subunit site. **f** Lipid site 4 at the base of TMD near the β-α subunit interface. **a**–**f** Subunits and lipids are colored as in Fig. 2.

the hydrocarbon tail of POPE inserting into the pocket where other modulators bind, including etomidate, propofol, and high concentrations of DMCM, diazepam, and zolpidem[24,42,43]. Photoaffinity labeling experiments have predicted neurosteroid binding near this ligand-binding pocket[34,41]. However, mutations in this pocket that inhibit etomidate potentiation did not affect neurosteroid potentiation or inhibition[41], consistent with our observation of a lipid tail occupying this site, rather than neurosteroids. The second outer-leaflet lipid, located between the M3 and M4 helices of the β subunits, was modeled as phosphatidylserine (POPS) based on the density features and local environment (Fig. 6, site 2). The POPS head group carboxylate orients to make electrostatic interactions with β2 R141 of the Cys-loop, while a

phosphate oxygen is positioned to make a hydrogen bond with the backbone nitrogen of β2 V278 in the M2-M3 loop (Fig. 6d). Both the Cys-loop and M2-M3 loops are central elements for channel gating in the receptor superfamily[36]. Previous studies have also shown lipidic composition at the ECD-TMD junction where the M2-M3 loop is housed can modulate receptor activity[48–50]. Together, we observe outer leaflet lipids poised to regulate channel activity through a site targeted by many allosteric modulators, but not the neurosteroids studied in this work, and through binding in a region central to the channel gating mechanism.

We observed two groups of inner leaflet lipid densities (Sites 3 and 4, Fig. 6e, f). Site 3 is adjacent to the M3 and M4 helices of the α

subunits. This site has been previously identified as either a binding site for PS in a GABA$_A$-GLIC chimera[19], or as a site for the lipid PIP2 in a full-length α1β3γ2 heteromeric receptor[51]. Our sulfated neurosteroid-bound GABA$_A$ receptor structures exhibit clear density for acyl chains indicative of a phospholipid at this site (Fig. 6e). The lipid density present in site 4 is adjacent to the β-M3 and α-M1 helices, surrounding the site where PAM neurosteroids bind (Site 4, Fig. 6a, f). We propose that this lipid stabilizes the binding of allopregnanolone and other PAM neurosteroids. We also note the presence of lipids in site 4 in other related structures from this superfamily[52,53]; this site appears to be a common locus for lipid-mediated stabilization of Cys-loop receptors.

## Discussion

Here we sought to delineate binding sites and allosteric mechanisms for neurosteroids with opposing activities. We found that the positive modulator allopregnanolone, a first in class therapeutic for post-partum depression, binds and acts through what is becoming the consensus PAM neurosteroid site. This site, at β-α interfaces in the TMD, is supported by photoaffinity labeling, and by experimental structures and robust mutagenesis results from several groups[13,15–17,19–21,30,31,41]. Binding of allopregnanolone to the α1β2γ2 receptor used in this study results in dramatic expansion of the extracellular end of the pore, in a manner similar to that seen with other PAMs that act through different sites, like propofol and etomidate[24]. This receptor conformation resembles a stable desensitized-like state, with GABA bound in the orthosteric sites, consistent with steady state expectations from electrophysiology in the presence of saturating GABA and allopregnanolone. These results place allopregnanolone in a category with other PAM neurosteroids like THDOC, alphaxalone, and pregnanolone[19–21], and provide a structural touchstone for therapeutic neurosteroids building off the success of allopregnanolone. An activated state structure would shed further light on PAM mechanism. Importantly, photoaffinity studies have highlighted additional sites for PAM neurosteroids beyond this single site identified through structural biology and mutagenesis[14,34,41,54]. Photoaffinity experiments can identify low occupancy sites, ones that structural biology experiments are likely to miss. Accordingly, we cannot rule out the possibility of additional sites for allopregnanolone outside this consensus site, however loss of allopregnanolone activity coincident with mutation of its major PAM site suggests that other sites would play minor signaling roles. Notably, a recent preprint on native GABA$_A$ receptors in rodents similarly identified only these two major allopregnanolone binding sites[25]. This solid understanding for PAM neurosteroid mechanism contrasts with the lack of consensus on how sulfated neurosteroids inhibit GABA$_A$ receptor activity.

The search for the mechanism(s) by which sulfated neurosteroids inhibit GABA$_A$ receptors has been going on for over 30 years[55–57]. Many results and discussions have pointed toward channel block. Notably, groups have found that the PAM and NAM neurosteroids act through different sites, with mutations that ablate PAM activity having no effect on NAM activity, which we also found[30,39]. Further, sulfated steroids must be added extracellularly to have an effect, suggesting they cannot partition into and traverse the membrane to access their site[44]. While PAM activity is strongly dependent on stereochemistry, NAM activity is, in comparison, much less sensitive to ligand geometry[39,40,58]. The only mutations that strongly diminish NAM activity are deep in the pore[40,44,59,60]. Lastly, inhibition by sulfated neurosteroids increases with increasing GABA concentration[37,45], suggesting use dependence, a hallmark of channel blockers. However, this channel-block basis of inhibition has been repeatedly discarded for several reasons, including a lack of strong voltage dependence, ambiguous competition with the famous pore blocker picrotoxin, and different kinetic effects compared to well-defined channel blockers. Additionally, the only direct structural information for a sulfated neurosteroid binding site[19] suggests it binds outside the ion channel. We discuss each of these arguments against channel block in turn.

Voltage dependence is often observed for charged channel blockers, in the case where its charge passes through the membrane field where the voltage gradient is greatest. Inhibition by PS and DHEAS have been shown by several groups to exhibit weak or no voltage-dependence[44,45,59]. Our structural results, combined with MD simulations, suggest stable pore binding only when the sulfate moiety orients toward the extracellular surface. The location of the negatively charged group approximately halfway across the membrane field is consistent with lack of a strong voltage dependence for binding at this site. Interestingly, sulfated neurosteroids also inhibit NMDA-type glutamate receptors, where they are use-dependent, not voltage-dependent, and are similarly proposed to act through a pore-block mechanism[61]. While voltage dependence can be a strong argument in support of a channel blocking mechanism, the absence of clear voltage dependence does not rule out this mechanism for inhibition. We suggest that the affinity for PS and DHEAS for the pore site stems from interactions of the hydrophobic steroid ring with hydrophobic pore lining residues, which, when rendered polar, cause a loss of inhibition for the sulfated neurosteroids. We further suggest that the charged sulfate group is important for dramatically increasing solubility and preventing membrane partitioning, and for stabilizing the anionic ligand inside an anion-selective channel. Still, the absence of voltage-dependence of block remains a weak point in our proposed pore block mechanism. Mutagenesis to render the pore more polar in regions we propose the steroid interacts could further test this mechanism.

Our structural results suggest that PS and DHEAS bind in a site overlapping with that where the classical channel blocker picrotoxin is known to bind in homomeric and heteromeric GABA$_A$ receptors, below the 9′ leucines[24,42,62]. Competition experiments using electrophysiology have explored this idea[44,45,63]. A recent study found that when the neurosteroids are applied with picrotoxin, there is a greater inhibition than when either the steroid or picrotoxin is applied alone[44]. These results were interpreted to suggest distinct, non-overlapping sites for the two classes of inhibitors. An alternative explanation is that in these whole-cell patch clamp experiments, there is a large population of receptors transitioning among different conformations, and picrotoxin is blocking some of them, while neurosteroids are blocking others, which results in the appearance of combined inhibition. Radioligand binding experiments to test PS and DHEAS competition against the channel blocker TBPS indeed observed monophasic competition and concluded physical overlap among the TBPS, picrotoxin, and sulfated neurosteroid sites[55,63]. These simple monophasic results differed from more complex inhibition between TBPS and allosterically acting ligands like pentobarbital, further supporting a conclusion that PS and DHEAS bind in the pore at a site overlapping with where picrotoxin binds.

The kinetic argument against a channel block mechanism is based on reported lack of effects on fast channel kinetics, and rather a more slowly developing block[59]. While this counterargument is speculative, the sulfated neurosteroids are both larger and much more hydrophobic than the fast-acting blockers like picrotoxin and TBPS, and intuitively would bind and dissociate more slowly. Another electrophysiology-based argument against pore block is that high concentrations of sulfated neurosteroids were found to cause incomplete block[63]. However, we and other groups[37,64] consistently see that at high sulfated neurosteroid concentrations the current rapidly decays, reversibly, to approximately zero.

While many of the counterpoints to proposed pore block mechanisms are based on functional measurements, there is one experimental structure from a homopentameric chimeric receptor with PS bound. This structure was obtained by x-ray crystallography at

3.0 Å resolution and was of a construct comprising the GABA$_A$ receptor α1 TMD and the ECD from GLIC, a pH-gated prokaryotic pentameric ligand-gated ion channel[19]. Crystallization conditions included detergent plus CHS, a water-soluble cholesterol derivative. The α1 subunit had a truncated ICD like that in our cryo-EM study, and a point mutation in the M2 helices reported to increase desensitization. This study concluded that PS binds at the interface of the α1 subunit and the inner leaflet of the lipid membrane. The electron density map reveals comparably strong density in the PAM neurosteroid site, which was not modeled, and density in the proposed PS site that could be explained by the added cholesteryl hemisuccinate. Mutations in the proposed PS site had modest effects on PS activity, which were comparable to the study's mutation of the stacking PAM site tryptophan. We were motivated to look at these details as recent structures from our group and others have modeled phospholipids in this earlier proposed PS site. We propose, based on the current structural data, and that sulfated neurosteroids cannot access their binding sites when applied intracellularly[44], that the earlier proposed PS site is not important for inhibition in the α1β2γ2 GABA$_A$ receptor.

Taken together, we define atomic level details for the interaction of allopregnanolone, an important therapeutic for postpartum depression, and provide experimental evidence for inhibition by sulfated neurosteroids exclusively via pore block. MD simulations and electrophysiology on mutant constructs complement the cryo-EM structures to sample dynamics and test the mechanistic hypotheses. We further discuss the rationale for a pore block mechanism in the context of decades of studies wherein the investigators appear tempted to draw this same conclusion but discarded it. Importantly, among the NAM neurosteroids, we focused here on well-studied sulfated steroids. Other non-sulfated inhibitory neurosteroids, like epi-allopregnanolone, may act through distinct mechanisms[41].

## Methods

### Receptor expression and purification

For structural studies, we expressed a tri-cistronic α1β2γ2 GABA$_A$ receptor construct that as described previously[24]. BacMam virus was generated from Sf9 cells (ATCC CRL-1711) and titrated[65]. HEK293S GnTI⁻ cells (ATCC CRL-3022) were grown in suspension at 37 °C with 8% $CO_2$ and transduced with multiplicities of infection of 0.5 when the cell density reached $3.5$–$4 \times 10^6$ cell per mL. To enhance protein expression, 1 mM sodium butyrate (Sigma-Aldrich) was added to the culture at the time of transduction and the temperature was reduced to 30 °C. After 72 h the cells were harvested by centrifugation. Cells were lysed using an Avestin Emulsiflex in 20 mM Tris pH 7.4 and 150 mM NaCl (TBS buffer) with 1 mM phenylmethanesulfonyl fluoride (PMSF; Sigma-Aldrich). At this point, 2 mM GABA was added in addition to the neurosteroids, each at the indicated concentration: 200 µM allopregnanolone (Sigma-Aldrich), 100 µM DHEAS, and 100 µM pregnenolone sulfate (Sigma-Aldrich). Lysed cells were centrifuged at $10,000 \times g$ for 20 min to pellet nuclei and unlysed cells. Membranes were collected via ultracentrifugation at $186,000 \times g$ for 2 h. Membranes were homogenized and solubilized in TBS containing ligands at the concentrations indicated above along with 40 mM *n*-dodecyl-β-maltoside (DDM, Anatrace) for 1 h at 4 °C. The solubilized membranes were then centrifuged for 40 min at $186,000 \times g$, and the supernatants were passed through Strep-Tactin XT Superflow affinity resin (IBA-GmbH). The resin was washed using TBS buffer that contained ligands, 0.01% (w/v) porcine brain polar lipids (Avanti), and 2 mM DDM and eluted in the same buffer that contained 50 mM biotin (Sigma-Aldrich).

### Receptor-nanodisc reconstitution

Salipro Biotech AB provided the saposin A expression plasmid. We modified a previously published protocol for reconstituting GABA$_A$ receptors into saposin-based nanodiscs[24,66]. The molar ratio of receptor, lipids, and saposin was 1:230:30. The α1β2γ2 receptors at a

concentration of approximately 15 µM were first mixed with porcine brain polar lipids for 10 min at room temperature. Saposin was then added and allowed to incubate for an additional 2 min. The reaction mixture was diluted ~10-fold with TBS to initiate reconstitution. Detergent was removed by adding Bio-Beads SM−2 (Bio-Rad) to a final concentration of 200 mg/ml while rotating overnight (~15 h) at 4 °C. The following day, BioBeads were removed, and the sample was concentrated for size-exclusion chromatography.

### Cryo-EM sample preparation

The reconstituted α1β2γ2 receptor was subsequently combined with 1F4 Fab at a ratio of 3:1 (w/w)[67]. After a 15-minute incubation period, the mixture was concentrated and passed through a Superose 6 Increase 10/300 GL column (GE Healthcare), which had been pre-equilibrated with ligands (200 µM allopregnanolone + 2 mM GABA; 100 µM DHEAS + 2 mM GABA or 2 mM pregnenolone sulfate + 2 mM GABA) in TBS. SEC fractions were assayed by fluorescence-detection size-exclusion chromatography monitoring intrinsic tryptophan fluorescence. Fractions that exhibited a single peak at the expected elution volume were collected, pooled, and concentrated to an A280 of 7–9. Prior to grid freezing, 0.5 mM fluorinated Fos-Choline-8 (Anatrace) was mixed with the sample to induce random orientations in the grid holes. Finally, 3 µL of the sample were placed on a glow-discharged gold R1.2/1.3 200 mesh holey carbon grid (Quantifoil), which was immediately blotted for 3 s at 100% humidity and 4 °C. Subsequently, the grids were plunge-frozen into liquid ethane using a Vitrobot Mark IV (FEI).

### Cryo-EM data collection and processing

Cryo-EM data were collected on a 300 kV Titan Krios Microscope (FEI), which was equipped with a K3 direct electron detector (Gatan) and a GIF quantum energy filter (20 eV) (Gatan). Super-resolution mode was used during data collection at the Pacific Northwestern Cryo-EM Center (PNCC). Detailed dataset-specific information can be found in Supplementary Data Table 1. A uniform workflow was used in Relion 3.1 to process all datasets[68]. Dose-fractionated images were gain-normalized, 2x Fourier binned, aligned, dose-weighted, and summed using MotionCor2[69]. Contrast transfer function (CTF) and defocus values were estimated using GCTF[70]. Cryolo 1.5.6[71] was used to pick particles for all datasets, which were then subjected to 2D classification. Good classes were selected for a second round of 2D classification, and an initial 3D model was generated using a few (5–7) classes. The initial model was then used for 3D classification. The best 3D classes were selected and low-pass-filtered to 40 or 50 Å to generate an improved initial model for 3D refinement. Particles from this refinement were polished. Due to high levels of disorder in the TMD of the γ-subunit observed in all datasets, focused 3D classification was performed on the γ-TMD after subtracting the signal from the rest of the receptor and nanodisc. Particles from the best classes were selected for 3D refinement followed by final round of particle polishing and an additional round of 3D refinement to generate the final maps. Relion 3.1[72] was used to estimate local resolution.

### Model building and refinement

The starting point for model building was the etomidate-bound α1β2γ2 receptor model (PDB: 6X3X). UCSF Chimera[73] was used to dock this model into the density map, which was then manually adjusted in Coot 0.9.4.1[74]. Ligand geometry restraints were built using the GRADE server[75]. The DHEAS-bound structure was initially built and served as a starting model for the allopregnanolone and pregnenolone sulfate complexes. Density was detected in the ECD α-γ benzodiazepine site in all three neurosteroid complex structures. Mutations showed that the TMD sites for allopregnanolone were the physiologically important ones; therefore, we modeled the ligand in only the TMD sites. Mutagenesis and pharmacology described in the results similarly ruled out this ECD site's relevance to NAM neurosteroid activity. Phenix 1.19-

4092[76] was used for global real space and B-factor refinement with stereochemistry restraints. Molprobity 4.5.2 was used to assess model quality. The PDBePISA[77] server was used to analyze subunit interfaces, and Hole2[78] was used to analyze pore radius. PROMALS3D[79] was used for sequence alignments, while UCSF Chimera 1.15[73] and UCSF ChimeraX 1.5[80] were used to generate figures. Structural biology software packages were compiled by SBGrid[81].

### Electrophysiology

We performed whole-cell voltage-clamp recordings from adherent HEK293S GnTI⁻ cells that were transiently transfected with either the tri-cistronic pEZT construct used for structural analysis or the WT construct. Cells were transfected with 0.2–0.5 µg of plasmid per well in a 12-well dish, and then maintained at 30 °C. On the day of recording, 1–3 days after transfection, the cells were re-plated onto a 35 mm dish and washed with bath solution comprising 140 mM NaCl, 2.4 mM KCl, 4 mM $MgCl_2$, 4 mM $CaCl_2$, 10 mM HEPES pH 7.3, and 10 mM glucose. Borosilicate pipettes were pulled and polished to an initial resistance of 2–4 MΩ. The pipette solution consisted of 150 mM CsCl, 10 mM NaCl, 10 mM EGTA, and 20 mM HEPES pH 7.3. The cells were clamped at −75 mV, and recordings were performed using an Axopatch 200B amplifier (Molecular Devices), sampled at 5 kHz, and low-pass-filtered at 2 kHz. The data were analyzed with pClamp 11 software (Molecular Devices). Solution exchange was accomplished using a gravity-driven RSC−200 rapid solution changer (Bio-Logic). Peak currents were plotted in bar graphs using GraphPad Prism 9 as mean ± standard deviation. Replicate values are listed in respective figure legends as $n$ = number of independent cells.

### Molecular dynamics

All-atom simulations in explicit solvent were deemed most appropriate to verify ligand poses and assess steady-state dynamics, given the relatively high precision and accuracy of atomistic interactions that can be captured compared to e.g. coarse-grained methods. Detailed dataset-specific information can be found in Supplementary Data Table 2. Atomic coordinates for the α1β2γ2 $GABA_A$ receptor determined by cryo-EM with different neurosteroid poses were used as starting models for MD simulations. The proteins were embedded in a bilayer of 400 1-palmitoyl-2-oleoyl-sn-glycero-3-phosphocholine (POPC) molecules and subsequently solvated with TIP3P water[82] and 150 mM NaCl in CHARMM-GUI[83], the simulation boxes were ~127 Å × 127 Å × 163 Å. The CHARMM36m forcefield[84] was used to describe the proteins. Parameters for the neurosteroids were generated by CGenFF[85] in CHARMM-GUI.

Simulations were performed using GROMACS 2020.5 or 2021.5[86] with temperature 300 K using the velocity-rescaling thermostat[87] and Parrinello−Rahman barostat[88]. The LINCS algorithm was used to constrain hydrogen-bond lengths[89], and the particle mesh Ewald method[90] was used to calculate long-range electrostatic interactions. Mutations were generated using the mutate-residue function in CHARMM-GUI during preparation. After each system was energy-minimized, sequential 10-ns equilibration steps were performed with gradual release of position restraints on heavy, backup, and C-alpha atoms. Four replicates each of 400–500 ns were simulated as final unrestrained production runs. Time courses for protein and ligand rmsd from their respective starting poses were monitored to verify equilibration, indicating that these systems converged appropriately within the timescale of atomistic simulations (e.g. Supplementary Fig. 3b–d, Supplementary Fig. 4a, b).

Root-mean-square deviations (rmsds) of ligands were calculated in VMD[91] while aligning on protein C-alpha atoms, and visualized with Matplotlib[92]. The upper-ECD spread was calculated based on the center-of-mass movement of the upper ECD, which includes residues 7–43, 62–99, 104–130, 154–176, and 194–208 of each β2 subunit, 10–45, 66–102, 107–133, 157–180, and 200–213 of each α1 subunit, and

25–57, 77–113, 119–145, 169–189, and 210–223 of the γ2 subunit. Neurosteroid interactions with protein were quantified using ProLIF 1.1.0[93]. Clustering analysis of neurosteroids was carried out using the clustering plugin in VMD (https://github.com/luisico/clustering), with total cluster number of 10 and rmsd cutoff of 1 Å.

Chloride permeation through the structure with GABA + Allo was calculated using the accelerated weight histogram (AWH) method[94]. Briefly, we applied one independent AWH bias and simulated for 50 ns with 16 walkers sharing bias data and contributing to the same target distribution. Each bias acts on the centre-of-mass z-distance between one central chloride ion and the Cα of residues β2-270, α1-275 and γ2-285, with a sampling interval across more than 95% of the box length along the z-axis to reach periodicity. To keep the solute close to the pore entrance, the coordinate radial distance was restrained below 10 Å by adding a flat-bottom umbrella potential. Permeation profiles for structures with GABA alone or GABA + propofol were as previously reported[24].

### Reporting summary

Further information on research design is available in the Nature Portfolio Reporting Summary linked to this article.

## Data availability

The data that support this study are available from the corresponding authors upon request. The cryo-EM maps have been deposited in the Electron Microscopy Data Bank (EMDB) under accession codes EMD-40503 (GABA + allopregnanolone); EMD-40462 (GABA + pregnenolone sulfate); and EMD-40506 (GABA + DHEAS). The atomic coordinates have been deposited in the Protein Data Bank (PDB) under accession codes 8SI9 (GABA + allopregnanolone); 8SGO (GABA + pregnenolone sulfate); and 8SID (GABA + DHEAS). Previously published structures compared in the study include: 6X3Z, 6X3X, 6X3V, and 6X3T. MD simulation trajectory, parameter files, and analysis scripts are available in Zenodo [10.5281/zenodo.7770004]. The source data underlying Figs. 3i, k, l, 5a–c, e–h, j, and Supplementary Figs. 3e, 5e–i, and 6h are provided as a Source Data file. Source data are provided with this paper.

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

## Acknowledgements

We thank J.J. Kim and S. Burke for guidance in sample preparation, J. Zhou, W. Chojnacka, H.H. Li, and S. Burke for feedback on the manuscript, and staff at the Pacific Northwest National Lab for cryo-EM data collection. We thank G. Akk for preliminary functional studies and discussion. Single-particle cryo-EM grids were screened at the University of Texas Southwestern Medical Center Cryo-Electron Microscopy Facility, which is supported by the CPRIT Core Facility Support Award RP170644. A portion of this research was supported

by NIH grant U24GM129547 and performed at the PNCC at OHSU and accessed through EMSL (grid.436923.9), a DOE Office of Science User Facility sponsored by the Office of Biological and Environmental Research. D.L. acknowledges a predoctoral fellowship from the National Institutes of Health (NIH, F31DA051176). C.F. was supported by grant FV-5.1.2-0523-19 from Stockholm University, and E.L. and R.J.H. by grants from the Swedish Research Council (2017-04641, 2019-02433) and Swedish e-Science Research Center. This project was supported by grants from the NIH (DA047325) and the Welch Foundation (I-1812) to R.E.H.

## Author contributions

D.L. performed the biochemistry, cryo-EM sample preparation, data processing, model building, refinement, structural analysis, and drafted the manuscript with C.M.N. and R.E.H. J.T. performed electrophysiology. C.M.N. screened the EM samples. C.F. and Y.Z. performed and analyzed MD experiments and wrote accompanying sections of the manuscript, with guidance from R.J.H. and E.L. All authors were involved in manuscript revision.

## Competing interests

The authors declare no competing interests.
