## [Peer Review File · Nature Communications]

Structural insights into opposing actions of neurosteroids on GABAA receptorsReviewers' Comments:

Reviewer #1:

Remarks to the Author:

Legesse et al present a structural study on neurosteroid modulation of the GABA(A) receptor examining the binding sites and structural effects of the PAM, allopregnanolone, and the NAMS, pregnenolone sulfate (PS) and DHEAS. These endogenous steroids are important modulators of the GABA(A) receptor and likely play a role in many physiologic processes. As these are the first structures of a non-chimeric GABA(A) receptor with neurosteroids, the findings along with the complementary MD simulations are an important contribution to the field. The results substantiate an established binding site for PAM neurosteroids, and provide interesting insights into the effects of bound allopregnanolone on the structure and dynamics of the receptor. The structural differences in the allopregnanolone-bound structure and the effects observed in the MD simulations suggest conformational changes that may underly allopregnanolone potentiation of GABA(A) receptor activity.

The study also presents structures with PS and DHEAS bound to the ion channel pore, suggesting a plausible mechanism by which sulfated neurosteroids inhibit/block the channel. The cryo-EM density of PS in the pore and the stability of PS in this site in the MD simulations provide convincing evidence that PS occupies this site. As discussed in the manuscript, a pore-block mechanism is consistent with some experimental findings such as the effect of the alpha1-V256S mutation. However, a simple pore-block mechanism is not entirely consistent with other findings related to PS and DHEAS block (see specific comments below). Also, while the authors present the results of mutations in the consensus site and benzodiazepine site, they do not perform mutagenesis at the site in the pore (other than referring to alpha1-V256S). Therefore, while the structural finding of PS and DHEAS in pore is an important contribution to the field, further work will be needed to test this hypothesis and to reconcile a pore-block mechanism with the functional data.

Below are specific comments for the authors to consider.

1) A recent study (reference 32) presented several findings regarding the inhibitory effects of PS and DHEAS that should be more carefully considered by the authors. The study showed that: 1) PS and DHEAS have a complex effect on GABA(A)R steady-state currents with an immediate peak inhibition followed by partial recovery of current, 2) there is a rebound current above the steady state current after removal of PS/DHEAS in the presence of GABA, and 3) PS/DHEAS do not completely inhibit GABA(A)R currents (a result that has been reported previously). These three findings, particularly finding #1, are not consistent with a simple pore-block mechanism. Further discussion is warranted.

2) The authors reference studies showing that alpha1-V256S reduced the rate of block by sulfated neurosteroids (reference 53). However, it is also important to consider the finding that mutations in the equivalent M2 position in beta2 and gamma2 subunits had no effect. This includes A252S in the beta2 subunit; beta2-A252 has comparable interaction frequency to alpha1-V256 with PS and DHEAS in the MD simulations. Also, it has been shown ([https://doi.org/10.1016/S0028-3908\(98\)00172-5](https://doi.org/10.1016/S0028-3908(98)00172-5)) that mutation of the 6' position in gamma2 (T271F) diminishes picrotoxin block, but has no effect on PS block. T271 also contacts PS and DHEAS in the MD simulations. To my knowledge, no other mutations in the PS/DHEAS pore site have been investigated, and therefore, the structures in this study provide a strong rationale for detailed functional studies into the effect of mutations at this site. It is also worth mentioning that alpha1-V256S, the only mutation that reduces PS/DHEAS block/inhibition, also reduces/abolishes the inhibitory effect of non-sulfated NAM neurosteroids including epi-allopregnanolone and allopregnanolone (DOI: 10.7554/eLife.55331, reference 35). While it is possible that these neurosteroids exert their NAM effects through binding to the pore, mutagenesis at photolabeled neurosteroid binding sites suggest otherwise (ref 35). Therefore, alpha1-V256S appears to reduce the NAM effects of non-sulfated neurosteroids through a down-stream effect.

3) Lines 313-314: It is worth mentioning based on reference 35 that other binding sites for non-

sulfated neurosteroids mediate the inhibitory effects of neurosteroids including allopregnanolone, which has both PAM and NAM activity. In a similar vein, the manuscript could better distinguish between sulfated neurosteroids, which are NAMs, as well as non-sulfated neurosteroids such as epiallopregnanolone and many other neurosteroid analogues, which also have NAM activity. For example, the paragraph starting in line 318 assumes that NAM neurosteroids equates to sulfated neurosteroids.

4) When stating that photolabeling (and mutagenesis/structural) studies have identified a consensus PAM neurosteroid binding site (lines 65, 301, 310), the authors have omitted the two primary photolabeling studies which identified these sites. These studies are: DOI: 10.1371/journal.pbio.3000157, and DOI: 10.1074/jbc.RA120.013452. The consensus site was also first photolabeled in the beta3 homopentamer in 2014 (DOI: 10.1124/mol.112.078410).

5) Is there any neurosteroid (allopregnanolone) or lipid density at the interface of subunits other than beta2/alpha1? A computational study (DOI: 10.1016/j.jsbmb.2018.04.012) suggested that allopregnanolone can occupy these other sites, and so this is worth noting/showing in the text or figures.

6) Are there any structural differences between the PS/DHEAS-bound structures and GABA-only structure? Unlike the allopregnanolone-bound structure, this comparison was not made. This is important considering a recent study proposing that PS stabilizes a non-conducting state that is not the same as the GABA-induced desensitized state (<https://doi.org/10.1124/molpharm.121.000385>).

7) The conformational differences in the allopregnanolone-bound structure and the movement of M3 observed in the MD simulations are quite interesting, since previous structures of neurosteroid-bound chimeric receptors do not show these differences. Supplementary Figure 4a illustrates the significant outward translation of the beta2 M3 in the allopregnanolone simulation. Was this observed in just one of the four replicates and in just one of the two alpha1 subunits? It is difficult to appreciate from the plots if this movement of M3 occurred in multiple replicates. Does this movement affect the interactions of allopregnanolone at its site? Also, does this movement lead to a significant widening of the pore at -2' in the simulations. It would also be helpful to clarify what is being measured in the y-axis of Fig. 3i. What is meant by "helical axis", and what distance is being measured for "M3 bottom expand"?

8) Fig. 2b, 4a-b, or other similar figures- While it is stated in the methods section that this is a whole-cell recording, it would be helpful to the reader if this was stated in the legend instead of "patch clamp electrophysiology".

9) Line 186. To my knowledge, there are no photolabeling studies describing the site for PS binding.

10) Lines 196-197. It would be helpful to the reader to cite the structural study being referenced in this statement.

11) In Supplementary Figure 5i, are relative peak currents being reported? Please clarify in legend.

12) Is the trace in Supplementary Figure 6g the EM construct? Please label.

13) Preferably, all electrophysiology results should be analyzed (peak currents, rate of desensitization, steady state currents or current after 2s agonist application) to show averaged results from all replicates, instead of just a single current trace.

14) In line 260, it is not clear that reference 40 relates to the statement. Also, I am not aware any studies looking at the impact of membrane lipids on the stability and activity of neurosteroids with the exception of a study on cholesterol (10.1038/sj.bjp.0704360).

15) In line 262, what does it mean for lipid densities to be “distal” from the neurosteroid site?

Reviewer #2:

Remarks to the Author:

The manuscript by Legesse et al reports on new cryo-EM structures of the $\alpha 1\beta 2\gamma 2$ GABAA receptor (GABAR) in complex with endogenous neurosteroids such as allopregnanolone (ALLO), pregnenolone sulphate (PS), and dehydroepiandrosterone (DHEAS). These structures solved in lipid nano-disks illuminate the binding sites and modes of potentiating and inhibiting neurosteroids with atomic resolution. The binding mode of ALLO corroborates the notion that potentiating neurosteroids bind to the transmembrane domain of the receptor at the interface of the $\beta 2$ - $\alpha 1$ subunits (i.e., intersubunit) and suggests that potentiation is mediated by a pore-dilation mechanism. The binding modes of PS and DHEAS provide evidence that inhibitory neurosteroids decrease the gating currents via a pore-block mechanism. Given the pharmacological relevance of GABARs that are targeted by benzodiazepines (e.g., diazepam, zolpidem and alprazolam), general anesthetics and barbiturates, and that allopregnanolone is the first FDA-approved allosteric drug targeting GABAR, this study is timely, pharmacologically relevant and of broad scientific interest. In addition, new mechanistic insights on allosteric potentiation emerge from this study and new arguments are raised to support the mechanism of allosteric inhibition by sulphated neurosteroids. However, some aspects of the study remain unclear and require, in my view, stronger support. Two major points are raised below. A list of minor points follows.

Major points:

1. The proposed mechanism of GABAR potentiation by neurosteroids involves the stabilization of a wider transmembrane pore when ALLO is bound. Although suggestive, this proposal is supported by structural and simulation data collected in the desensitized state (“this receptor conformation resembles a stable desensitized-like state” on page 12), which is likely to be non-conductive. As such, I would argue that the proposed mechanism of potentiation remains speculative. Since other mechanistic hypotheses for PAM modulation exist, e.g., those relying on differential ligand-binding affinities for conductive versus non-conductive state (10.1016/j.mam.2021.101044), stronger evidences are required to support the pore-dilation mechanism. For instance, if the GABAR structure with ALLO bound is relevant for ion conductance, computational electrophysiology and/or permeation free energy profiles of GABA-only versus GABA+ALLO might provide additional insights. Moreover, the conformational change leading to pore dilation in the presence of ALLO should be presented in greater detail. It is unclear, for instance, whether the tilting/twisting of the β -M2 helix associated to pore dilation (Fig.S4) is a genuine cryo-EM result or a simulation prediction.

2. The mechanism for inhibition by sulphated neurosteroids is extensively discussed in relation to previous literature proposals based on functional and structural data. However, since the controversy is long standing (>30 years), much caution and very strong arguments are needed. For instance, I do not fully understand the counter argument to justify the lack of strong voltage dependence of inhibition by PS and DHEAS on page 11, which was used in the past to discard pore blocking. Also, the pore-blocking mechanism proposed here does not really explain the modest but existing effect of mutations in the region corresponding to the PS-binding site highlighted by X-ray crystallography of the GABA/GLIC chimera (10.1038/nsmb.3477). To resolve the controversy, the authors could expand/clarify their arguments and strengthen their conclusion.

In this context, I was wondering whether detection of rectification in the I/V curve in the presence of PS or DHEAS would be a way to provide stronger evidence of pore blocking as established in glutamate-gated NMDA receptors (10.1016/0896-6273(95)90049-7) or the inward-rectifier K⁺ channels (10.1146/annurev.physiol.66.032102.150822); since sulphate neurosteroids are charged, I would expect much more efficient inhibition by pore block in one voltage direction relative to the other.

Minor points:

- Introduction: The Introduction is very short and much introductory information is given in the Results section. The authors might want to consider changing this organization, prepare the ground since the beginning, and focus only on the experimental evidence later.
- Page 3: the sentence "to contrast GABA receptor binding..." reads odd.
- Page 4: the authors refer to an $\alpha 1\beta 2\gamma 2$ GABA receptor construct. What was modified in the construct relative to the physiological receptor? This important information should be given in the Main Text and not only in the Methods.
- Page 5: "hydrophobic stacking interactions". Not sure it makes sense to talk about hydrophobic interactions in a non-polar environment like a lipid bilayer...
- Page 5: the functional work showing the role of W246 could be described more explicitly in the Main Text (which mutations were done?). The same for L301 for which no Ref is given.
- Figures: Fig.2-3-4-S5-S6 are rather dark and way too small so details (including residue numbering and interactions) are difficult to appreciate.
- Page 7: the sentence starting with "We conclude..." comes too early and should be moved later in the story.
- Page 7: why $\alpha W245$ and not $\alpha W246$ is discussed here as done in Fig.2?
- Page 7: "presence of sticky steroid binding site". Since electron-density was observed in cryo-EM, which steroid could be possibly present during sample preparation?
- Page 7: "Supp Fig5e-h" should be "Supp Fig5i-j"
- Page 7: "neither mutations in the α - γ interface... affected inhibition by PS". How significant are the differences between H102R and WT observed in Fig.S6?
- Page 8: the introduction paragraph on how lipids help fixing neurosteroids to GABAR as part of their binding site is misleading since lipid densities are discussed later in the subsection in competition with neurosteroid binding.
- Page 9: "GABA:ELIC chimera" should be "GABA:GLIC chimera" as discussed on page 12
- Page 11: Add Ref to the work of Laverty et al (10.1038/nsmb.3477) on the first sentence.
- Page 12: CHS should be spelled in full

We thank both peer reviewers for their constructive comments, suggestions, and questions. Below we provide point-by-point responses in blue font.

Reviewer #1

Legesse et al present a structural study on neurosteroid modulation of the GABA(A) receptor examining the binding sites and structural effects of the PAM, allopregnanolone, and the NAMs, pregnenolone sulfate (PS) and DHEAS. These endogenous steroids are important modulators of the GABA(A) receptor and likely play a role in many physiologic processes. As these are the first structures of a non-chimeric GABA(A) receptor with neurosteroids, the findings along with the complementary MD simulations are an important contribution to the field. The results substantiate an established binding site for PAM neurosteroids, and provide interesting insights into the effects of bound allopregnanolone on the structure and dynamics of the receptor. The structural differences in the allopregnanolone-bound structure and the effects observed in the MD simulations suggest conformational changes that may underly allopregnanolone potentiation of GABA(A) receptor activity.

The study also presents structures with PS and DHEAS bound to the ion channel pore, suggesting a plausible mechanism by which sulfated neurosteroids inhibit/block the channel. The cryo-EM density of PS in the pore and the stability of PS in this site in the MD simulations provide convincing evidence that PS occupies this site. As discussed in the manuscript, a pore-block mechanism is consistent with some experimental findings such as the effect of the alpha1-V256S mutation. However, a simple pore-block mechanism is not entirely consistent with other findings related to PS and DHEAS block (see specific comments below). Also, while the authors present the results of mutations in the consensus site and benzodiazepine site, they do not perform mutagenesis at the site in the pore (other than referring to alpha1-V256S). Therefore, while the structural finding of PS and DHEAS in pore is an important contribution to the field, further work will be needed to test this hypothesis and to reconcile a pore-block mechanism with the functional data.

We thank the reviewer for their enthusiasm about the important aspects of the contribution to the field and address comments about conformational state and pore mutagenesis below.

Below are specific comments for the authors to consider.

1) A recent study (reference 32) presented several findings regarding the inhibitory effects of PS and DHEAS that should be more carefully considered by the authors. The study showed that: 1) PS and DHEAS have a complex effect on GABA(A)R steady-state currents with an immediate peak inhibition followed by partial recovery of current, 2) there is a rebound current above the steady state current after removal of PS/DHEAS in the presence of GABA, and 3) PS/DHEAS do not completely inhibit GABA(A)R currents (a result that has been reported previously). These three findings, particularly finding #1, are not consistent with a simple pore-block mechanism. Further discussion is warranted.

We found this study to be particularly thoughtful and we agree with its major conclusions: that PS and DHEAS share a common site, and that these inhibitors stabilize a receptor state that has high affinity for neurotransmitter, like a desensitized state, but is distinct from the classical desensitized state with only neurotransmitter bound. We respond here to your specific points 1-3 as a-c:

a: PS and DHEAS have a complex effect on steady state currents. Yes, we agree, for example in Fig. 2b of the cited study (was ref 32, PMID 34853153). At low concentrations, PS gives a non-stable partial block. However, at high concentrations, PS gives a ~complete and stable block, consistent with what we see in our presented electrophysiology experiments. We do not see this variable block, but we did not test the low end of the concentration range (0.3-3 μ M) as we were focusing mainly on concentrations we could correlate with the cryo-EM samples. A possible explanation for the slowly equilibrating block is the slow solution exchange in TEVC experiments vs the whole HEK cell patch clamp experiments we performed. Another possible explanation (not exclusive to the first) is that PS can bind to and block multiple conformational states- for example, an activated state and a desensitized state, or an intermediate, and binds to one with higher affinity than the other. Slow conformational equilibration may explain the slow change in amount of block in the low concentration range of PS. These speculative ideas are in line with those discussed in the reference, related to their Figure 3 results.

b: Rebound current above the steady state current upon PS/DHEAS washout. This result is nicely illustrated in ref 32 Fig. 3c. This finding reminded us of studies on picrotoxin, the prototypical GABA_A channel blocker. Trevor Smart's group used this phenomenon to study which conformational state(s) PTX binds to (PMID 25891813) in GABA_A and glycine receptors. We did something similar in PMID 32879488 (see its Supplementary Figure 3c,d). PTX in this case both blocks the pore and shifts the conformational equilibrium away from a/the desensitized state, speeding recovery from desensitization. It looks, from ref 32, like PS and DHEAS are doing the same thing- which, drawing the parallel to picrotoxin, is not an argument against a pore block mechanism. Indeed, these large rebound currents upon PTX and sulfated neurosteroid washout are arguments in favor of them both being pore blockers.

c: Incomplete block has been reported previously, however the concentrations were not high. Complete block has also been reported previously, in studies we cite in the submitted manuscript. We also show ~complete block in Fig 4. Ref 32 estimates 99% block of steady state currents and 97% block of peak currents at saturating neurosteroid concentrations. Slightly smaller fractional block of peak currents makes sense in the model we suggest where the channel must open in order for the blocker to bind. Achieving absolute complete block would only be theoretically possible with a covalent inhibitor and an exceptionally fast on rate. In our view, 99% block as estimated in the reference 32 is equivalent to a complete block for the purpose of assessing where an antagonist binds.

An important overall point we take from this reviewer's comment is that we needed to be clearer in our language in the text: we are not arguing that PS and DHEAS bind to the exact same state(s) stabilized by GABA. Rather, we propose that the major source of inhibition by these neurosteroids comes from binding in the pore- to one or more conformations. We have adjusted the text in the manuscript to be clearer about the conformational state to which the sulfated, inhibitory neurosteroids bind. For example, when we are introducing the activity of neurosteroids, which increase the rate of current decay in the presence of agonist, we no longer call this simply desensitization, but say that they accelerate apparent desensitization. We also performed a new analysis of pore conformations, shown in Supplementary Fig. 7, illustrating how PS and DHEAS stabilize pore conformations distinct from both the GABA-bound and GABA + allopregnanolone complexes, consistent with the PMID 34853153 findings.

2) The authors reference studies showing that alpha1-V256S reduced the rate of block by sulfated neurosteroids (reference 53). However, it is also important to consider the finding that mutations in the equivalent M2 position in beta2 and gamma2 subunits had no effect. This includes A252S in the beta2 subunit; beta2-A252 has comparable interaction frequency to alpha1-V256 with PS and DHEAS in the MD simulations. Also, it has been shown ([https://doi.org/10.1016/S0028-3908\(98\)00172-5](https://doi.org/10.1016/S0028-3908(98)00172-5)) that mutation of the 6' position in gamma2 (T271F) diminishes picrotoxin block, but has no effect on PS block. T271 also contacts PS and DHEAS in the MD simulations. To my knowledge, no other mutations in the PS/DHEAS pore site have been investigated, and therefore, the structures in this study provide a strong rationale for detailed functional studies into the effect of mutations at this site. It is also worth mentioning that alpha1-V256S, the only mutation that reduces PS/DHEAS block/inhibition, also reduces/abolishes the inhibitory effect of non-sulfated NAM neurosteroids including epi-allopregnanolone and allopregnanolone (DOI: 10.7554/eLife.55331, reference 35). While it is possible that these neurosteroids exert their NAM effects through binding to the pore, mutagenesis at photolabeled neurosteroid binding sites suggest otherwise (ref 35). Therefore, alpha1-V256S appears to reduce the NAM effects of non-sulfated neurosteroids through a down-stream effect.

Thank you for bringing up the results on other mutations in the pore that do not cause apparent changes in PS and/or DHEAS inhibitory activity or potency. On the topic of β 2-A252S, indeed the first paper on this mutant (PMID 11313438) reported no significant change in PS inhibition. However, as discussed in PMID 29447845 (Sandra Seljeset first author, Smart lab), two subsequent papers (PMIDs 17054655, 17239367) show a large loss of PS inhibition with this β 2 mutation. Helpfully, the PhD thesis from Sandra Seljeset, which we now cite, is easily searchable online (<https://discovery.ucl.ac.uk/id/eprint/1537304/>), and shows very nice recordings from 2' mutants, finding that the beta3 2' alanine to serine mutation causes a ~50-fold decrease in IC50 for PS. This loss of inhibitory activity in β and α 2' mutants is consistent with the instability for PS and DHEAS in our MD simulations of these mutants. On the topic of γ 2, this mutant was only reported, to our knowledge, in the PMID 11313438 study. The mutant studies in this paper (for α , β , and γ subunits) show only exemplar single channel records, which are only 2 seconds in length. Earlier in the paper, detailed single channel analysis is performed to study GABA kinetics. However, in the mutants, to test the effects of PS,

no statistical analysis is performed, and no quantitative data are presented. We suggest that the later studies showing that the 2' position is indeed very important in PS inhibition should be weighed as more robust given that they show data from multiple recordings with statistical analysis.

Next, we discuss the 6' mutation to phenylalanine, in the $\gamma 2$ subunit, that the reviewer raised. This relates to the topic we discussed in the manuscript, where picrotoxin has been suggested to either overlap with or not overlap with a pore block site for sulfated neurosteroids. We thank the reviewer for drawing our attention to this paper (PMID 10218867), which we had missed. This paper compares, in each of 3 figures, WT GABA_A receptor vs. a double-mutant receptor where the gamma2 subunit has both T271F (6') and T277F (13'). Figure 1 is a single panel fitted response comparing WT vs. the mutant receptor at 4 concentrations, finding that PTX does not inhibit the mutant but does potently inhibit WT. Importantly, they show no recordings, only fitted data. Figures 2 and 3 show the same results but for PS and DHEAS inhibition respectively, with the presentation as follows. Panels A and B show example TEVC recordings from WT vs. the mutant that reveal a major loss of response to GABA in the mutant construct- 10x smaller currents. PS does not appear to be substantially affected by this mutation. Panel C presents dose-response fitting of PS inhibition of WT vs the mutant. Figure 3 is like Fig 2 but for DHEAS. Here the example recordings show a major change in inhibition by DHEAS- apparently in kinetics- for the mutant compared to WT, however from the way these data are plotted in C, this change is not obvious in terms of IC50. We have a few thoughts about the implications of this study. First, it is important to consider that when one makes mutations in the M2 helices, and in the case of this study replaces a small polar residue with a bulky hydrophobic residue in a pore lining position, it is very likely to alter fundamental channel activity. Interpreting the results in a simple way, in the absence of thorough controls, is not possible. The channel structure may very well be distorted from what WT looks like. Nonetheless, a simple explanation for why PTX may be so dramatically affected by this 6' mutation is that 6' is an essential polar residue for coordinating the caged oxygens in the pore blocker. This is not the case for PS and DHEAS. Indeed, the MD simulations from our study show relatively rare interactions at 6' compared to residues at the 9', 5', 2', or -2' positions, consistent with the $\gamma 2$ -T271F mutation not having a major effect.

On the topic of expanding this study to look at pore mutants by electrophysiology, a concern with making many mutants in the pore is affecting basic channel function (nicely illustrated by the 6' mutation study discussed in the preceding paragraph, where a large loss of GABA activity was shown). This would be a major undertaking- to test several mutants, get agonist EC50s, show that other channel properties are not grossly affected (conductance, selectivity, kinetics). A thorough mutagenesis and electrophysiology study would certainly be a way to further interrogate the proposed mechanism, however we feel this is beyond the scope of the current structure-dynamics project.

Differences between sulfated and non-sulfated NAM neurosteroids:

We chose to focus, in this study, on sulfated NAM neurosteroids, as there was a precedent in the structural biology literature. The original Ref 35 (PMID: 32955433), highlighted in this reviewer comment, addresses non-sulfated NAM neurosteroids, which we did not study. We intentionally did not dive into the distinct question of how non-sulfated NAMs act. We agree, however, that sulfated and non-sulfated neurosteroids are likely to act at least in part through different inhibitory sites. We favor the idea that ligand solubility is a major driver of the differences. Sulfated neurosteroids will not penetrate deeply into a lipid bilayer, so they cannot inhibit through the consensus PAM site. Their concentrations in aqueous solution will be comparably high, so they can act as effective pore blockers. DHEA and epi-allopregnanolone, in contrast, will partition selectively into the lipid bilayer, where their local concentrations will be much higher than in the aqueous solution- for this reason, they are unlikely to function as potent/efficacious pore blockers. In exploratory laboratory tests, we found that DHEA (and not DHEAS) inhibition was diminished by a combined set of three mutations (the same as in our study) in the 'consensus' PAM site we and others visualized at β - α interfaces. Moreover, as shown at right, simulations of this system showed DHEA to be stable in this consensus PAM site (relative to DHEAS, as shown in the manuscript Supplementary Fig. 5g),

and to be disrupted by Q242L. It would be interesting to test epi-allopregnanolone with a similar set of mutations, as part of a separate structure-function study on non-sulfated NAM mechanisms.

We have adjusted the manuscript language to be explicit about our focus on sulfated neurosteroids in the current study. We title the relevant section: "Structural basis of sulfated neurosteroid inhibition." We start this section with: "Sulfated neurosteroids..." In the discussion, where we introduce which NAM neurosteroids we are going to discuss, we explicitly call them sulfated neurosteroids (lines 317-319).

3) Lines 313-314: It is worth mentioning based on reference 35 that other binding sites for non-sulfated neurosteroids mediate the inhibitory effects of neurosteroids including allopregnanolone, which has both PAM and NAM activity. In a similar vein, the manuscript could better distinguish between sulfated neurosteroids, which are NAMs, as well as non-sulfated neurosteroids such as epiallopregnanolone and many other neurosteroid analogues, which also have NAM activity. For example, the paragraph starting in line 318 assumes that NAM neurosteroids equates to sulfated neurosteroids.

Thank you for raising this important distinction. We have adjusted the text to more explicitly state that we are focused on sulfated NS and know that there are other NAM neurosteroids that are not sulfated. It makes good sense that through differential membrane partitioning the inhibitory mechanisms would be different. The final sentence in the main text is new:

"Other non-sulfated inhibitory neurosteroids, like epi-allopregnanolone, may act through distinct mechanisms(ref)." (Ref is PMID 32955433 from the Evers lab.)

4) When stating that photolabeling (and mutagenesis/structural) studies have identified a consensus PAM neurosteroid binding site (lines 65, 301, 310), the authors have omitted the two primary photolabeling studies which identified these sites. These studies are: DOI: 10.1371/journal.pbio.3000157, and DOI: 10.1074/jbc.RA120.013452. The consensus site was also first photolabeled in the beta3 homopentamer in 2014 (DOI: 10.1124/mol.112.078410).

Thank you, we have incorporated these references.

5) Is there any neurosteroid (allopregnanolone) or lipid density at the interface of subunits other than beta2/alpha1? A computational study (DOI: 10.1016/j.jsmb.2018.04.012) suggested that allopregnanolone can occupy these other sites, and so this is worth noting/showing in the text or figures.

We do not observe lipid or neurosteroid-like density at the equivalent of the PAM site in other subunit interfaces.

6) Are there any structural differences between the PS/DHEAS-bound structures and GABA-only structure? Unlike the allopregnanolone-bound structure, this comparison was not made. This is important considering a recent study proposing that PS stabilizes a non-conducting state that is not the same as the GABA-induced desensitized state (<https://doi.org/10.1124/molpharm.121.000385> ref.32)

Thank you for bringing up conformational differences between the GABA-only vs. GABA + sulfated neurosteroid complex structures. Indeed, there are differences consistent with the findings of that paper on the sulfated neurosteroids binding to a conformation distinct from a desensitized state stabilized by GABA alone. The overall rmsd's results from global superpositions are 0.60 and 0.58 for PS vs. GABA-only and DHEAS vs. GABA-only, respectively. The differences are minor in the ECDs, where agonist is bound, stabilizing that domain in a clearly desensitized-like conformation. The differences are concentrated in the TMD; rmsd's for this domain are 0.78 and 0.78 for the two complexes vs. GABA-only. As mentioned in response to Question 1, we have added a new Supplementary Figure 7 comparing GABA vs. PS vs. DHEAS vs. allopregnanolone complex pore conformations. We refer to this finding in relation to the study referenced in your question near the end of the results section on pore binding of sulfated neurosteroids.

7) The conformational differences in the allopregnanolone-bound structure and the movement of M3 observed in the MD simulations are quite interesting, since previous structures of neurosteroid-bound chimeric receptors do not show these differences. Supplementary Figure 4a illustrates the significant outward translation of the beta2 M3 in the

allopregnanolone simulation. Was this observed in just one of the four replicates and in just one of the two alpha1 subunits? It is difficult to appreciate from the plots if this movement of M3 occurred in multiple replicates. Does this movement affect the interactions of allopregnanolone at its site? Also, does this movement lead to a significant widening of the pore at -2' in the simulations. It would also be helpful to clarify what is being measured in the y-axis of Fig. 3i. What is meant by “helical axis”, and what distance is being measured for “M3 bottom expand”?

We appreciate both reviewers' clarifying questions regarding this result. Beginning with the last question about the measurement of M3 movement, the y-axis tracks the displacement of the C α atom of the innermost pore-oriented residue in M3 (β 2-N303, α 1-N308, or γ 2-H318) relative to its starting position. Regarding the extent of this dynamic feature, relative mobility of the inward-facing portion of M3 in the presence versus absence of allopregnanolone was evident specifically in the β 2 subunit distal to γ 2 (chain C), not in the other four subunits:

Regarding the reproducibility of this effect, elevated mobility of this portion of β 2(chain C) was apparent in all replicates, as shown below in the bottom time courses below (colored by chain, β 2(chain C) in dark blue). The effect was most dramatic in replicate 1, where the region sampled deviations >7 Å. In simulations of the GABA-only system shown in the top time courses, the mobility of this portion of β 2(chain C) was similar to or lesser than that of γ 2 (yellow), deviating <3 Å in all but one replicate:

Regarding corresponding motions in allopregnanolone itself, we did not observe substantial deviation of the steroid in association with chain C (ALP2, dark blue) relative to chain A (ALP1, light blue), except modestly in replicate 1, where the dramatic outward translocation of M3 shifted several contacting residues. Since this motion was only observed once, and in association with a possibly rare motion of the protein, we conservatively summarize that the steroid itself maintains a largely stable position in the TMD site:

Regarding corresponding changes in the desensitization gate, as shown in the time courses below (colored by replicate), the radius at -2' was more variable in simulations with GABA+ALP (bottom) than with GABA alone (top)—though it was generally constricted rather than widened in the presence of the drug. Indeed, the replicate with the largest translocation of M3 (GABA+ALP, replicate 1, blue) corresponded to the tightest constriction at -2', from 2.0 Å to as low as 0.5 Å. Although it is possible this flexibility in the inward-facing pore could be associated with relative accessibility of the open state, the absence of direct structural data for the latter limits interpretation of its mechanistic impact.

Taken together, these results demonstrate consistent mobilization of the inward-facing portion of $\beta 2$ (chain C) in the presence of allopregnanolone. However, it is not straightforward to rationalize why no such mobility is observed in $\beta 2$ (chain A), nor what its direct impact might be on pore state or coupling. Since these reviews are public, we provide these analyses here for purposes of transparency and the public record. However, to focus the manuscript more specifically on dynamic features previously implicated in channel gating, and remove points of confusion voiced by the reviewers, we have removed the text and figure panels (3i, S4a–b) describing these observations, until our ongoing studies may clarify their relevance.

8) Fig. 2b, 4a-b, or other similar figures- While it is stated in the methods section that this is a whole-cell recording, it would be helpful to the reader if this was stated in the legend instead of “patch clamp electrophysiology”.

We have revised the legends accordingly.

9) Line 186. To my knowledge, there are no photolabeling studies describing the site for PS binding.

Absolutely, we apologize for the error, and have removed the photolabeling part of this sentence.

10) Lines 196-197. It would be helpful to the reader to cite the structural study being referenced in this statement.

Done.

11) In Supplementary Figure 5i, are relative peak currents being reported? Please clarify in legend.

Correct. We have clarified that these are relative peak currents in the legend in the revised manuscript.

12) Is the trace in Supplementary Figure 6g the EM construct? Please label.

Correct, and we now define that the background construct for all these functional measurements is the EM construct.

13) Preferably, all electrophysiology results should be analyzed (peak currents, rate of desensitization, steady state currents or current after 2s agonist application) to show averaged results from all replicates, instead of just a single current trace.

In the current study, we used electrophysiology mainly to make qualitative assessments of activity and binding sites. We are not trying to make quantitative statements about kinetics, or which state is being stabilized based on electrophysiology. Accordingly, we do not see how detailed analysis of all recordings would benefit the study. More specifically, we use whole cell patch clamp electrophysiology to illustrate the PAM activity of allopregnanolone; this behavior has been studied in depth by many labs. We use the same approach to similarly illustrate the inhibitory activity of sulfated neurosteroids, again work that has been done in depth by several other groups. Lastly, we use mutagenesis and pharmacology to rule out the inhibitory activity from sulfated neurosteroids arising from binding in specific locations. For this study, we are not interested in how these mutations affect subtleties of channel gating. Having said that, Rev 2 asked a question about details of mutant electrophysiology that motivated us to add a bar graph showing all individual recordings, mean and SD; this new analysis is shown below and is featured as a new panel in Supplementary Figure 6.

14) In line 260, it is not clear that reference 40 relates to the statement. Also, I am not aware any studies looking at the impact of membrane lipids on the stability and activity of neurosteroids with the exception of a study on cholesterol (10.1038/sj.bjp.0704360).

We agree about the relevance of ref 40. Thank you for highlighting this study on cholesterol showing dependence of PAM and NAM neurosteroid activity on the lipid environment. We replaced the original reference there with this cholesterol study (now ref 46, PMID 11704651).

15) In line 262, what does it mean for lipid densities to be “distal” from the neurosteroid site?

What we meant is that there are lipids that appear to directly contact bound PAM neurosteroids, and lipids more distant from this site, which do not directly contact the ligand. The revised sentence is:

“The cryo-EM structures revealed several well-ordered lipid densities positioned to form a component of the neurosteroid binding sites, as well as lipid densities distal from the PAM neurosteroid site likely important for receptor stability (Figs. 2, 3, and 6).”

Reviewer #2

The manuscript by Legesse et al reports on new cryo-EM structures of the $\alpha 1\beta 2\gamma 2$ GABAA receptor (GABAR) in complex with endogenous neurosteroids such as allopregnanolone (ALLO), pregnenolone sulphate (PS), and dehydroepiandrosterone (DHEAS). These structures solved in lipid nano-disks illuminate the binding sites and modes of potentiating and inhibiting neurosteroids with atomic resolution. The binding mode of ALLO corroborates the notion

that potentiating neurosteroids bind to the transmembrane domain of the receptor at the interface of the $\beta 2$ - $\alpha 1$ subunits (i.e., intersubunit) and suggests that potentiation is mediated by a pore-dilation mechanism. The binding modes of PS and DHEAS provide evidence that inhibitory neurosteroids decrease the gating currents via a pore-block mechanism. Given the pharmacological relevance of GABARs that are targeted by benzodiazepines (e.g., diazepam, zolpidem and alprazolam), general anesthetics and barbiturates, and that allopregnanolone is the first FDA-approved allosteric drug targeting GABAR, this study is timely, pharmacologically relevant and of broad scientific interest. In addition, new mechanistic insights on allosteric potentiation emerge from this study and new arguments are raised to support the mechanism of allosteric inhibition by sulphated neurosteroids. However, some aspects of the study remain unclear and require, in my view, stronger support. Two major points are raised below. A list of minor points follows.

We thank this reviewer for their enthusiasm about the timeliness and broad relevance of the study, and for their constructive suggestions about how to further clarify and strengthen the manuscript.

Major points:

1. The proposed mechanism of GABAR potentiation by neurosteroids involves the stabilization of a wider transmembrane pore when ALLO is bound. Although suggestive, this proposal is supported by structural and simulation data collected in the desensitized state (“this receptor conformation resembles a stable desensitized-like state” on page 12), which is likely to be non-conductive. As such, I would argue that the proposed mechanism of potentiation remains speculative. Since other mechanistic hypotheses for PAM modulation exist, e.g., those relying on differential ligand-binding affinities for conductive versus non-conductive state (10.1016/j.mam.2021.101044), stronger evidences are required to support the pore-dilation mechanism. For instance, if the GABAR structure with ALLO bound is relevant for ion conductance, computational electrophysiology and/or permeation free energy profiles of GABA-only versus GABA+ALLO might provide additional insights. Moreover, the conformational change leading to pore dilation in the presence of ALLO should be presented in greater detail. It is unclear, for instance, whether the tilting/twisting of the β -M2 helix associated to pore dilation (Fig.S4) is a genuine cryo-EM result or a simulation prediction.

Thank you for the clarifying question. A challenge has been that at equilibrium, the default condition for structural biology, agonist + PAM nearly always settles into a desensitized (or similar) state. This finding is consistent with predictions from electrophysiology experiments at very long time points. As such, we must infer mechanisms of potentiation from a desensitized or desensitized-like state. There is an emerging consensus among pentameric channels, especially GABA_A receptors and glycine receptors, that PAMs that bind in the TMD exert their potentiating effects by causing pore dilation in the upper half of the pore. This region comprises the activation gate, which is strongly hydrophobic. Ligands that stabilize a more widely open activation gate at the 9' position and above, and keep this gate open, would logically favor an activated state over a resting state. Indeed, as the reviewer suggests, the allo-bound structure has a closed desensitization gate, which would render computational electrophysiology uninformative. However, we quantified pore hydration at the activation gate (9') in Figure 3; we show here that allo (ALP) dramatically increases the number of waters in this location (Fig. 3i), reflecting a destabilized activation gate.

To further quantify the thermodynamic effect arising from pore widening with the PAM, we calculated the free-energy profile for chloride ion permeation along the axis of the structure with GABA + allopregnanolone. As seen at right, the resulting profile (red) showed a lowering of the energy barrier to permeation around the 9' hydrophobic gate (1.5 nm along the y-axis) relative to the structure with GABA alone (blue), similar to the lowered barrier previously reported for the structure with GABA + propofol. We have now added this profile as Fig. 3j, and added corresponding text to Methods and Results.

Regarding structural and dynamic changes in the extended TMD, we have clarified that the tilting/twisting depicted in Fig. 3b represents cryo-EM data. As described above in response to reviewer 1 – question 7, we also have streamlined our

description of allopregnanolone effects by removing reference to mobilization of the beta subunit in MD simulations.

2. The mechanism for inhibition by sulphated neurosteroids is extensively discussed in relation to previous literature proposals based on functional and structural data. However, since the controversy is long standing (>30 years), much caution and very strong arguments are needed. For instance, I do not fully understand the counter argument to justify the lack of strong voltage dependence of inhibition by PS and DHEAS on page 11, which was used in the past to discard pore blocking. Also, the pore-blocking mechanism proposed here does not really explain the modest but existing effect of mutations in the region corresponding to the PS-binding site highlighted by X-ray crystallography of the GABA/GLIC chimera (10.1038/nsmb.3477 (ref 15). To resolve the controversy, the authors could expand/clarify their arguments and strengthen their conclusion.

Our general argument is that pore block explains more of the results than any other mechanism. We bring in additional mutagenesis from the literature, in response to a reviewer 1 question (now cited in the text), supporting a pore block mechanism. Voltage dependence can be used to argue for a pore block mechanism but absence of voltage dependence cannot be used to rule out pore block. The interaction of an antagonist with its site can be stabilized through numerous types of interactions. We suggest that interactions of the hydrophobic steroid ring with hydrophobic residues lining the pore are more important determinants than the charged sulfate. What the charged sulfate adds is 1) much improved solubility compared to the lipid-partitioning, non-sulfated neurosteroids, and 2) preferential binding to an anion-selective channel. We suggest that these are the drivers of affinity for PS and DHEAS in the pore, with voltage playing a minor (or no) role. Consistent with this idea, PS has recently been shown to be a voltage-independent blocker of NMDA receptors (PMID 26086919). We have expanded this section of the discussion accordingly.

Regarding the crystallographic study with PS. First, we note that extrapolation is likely to be imperfect from a homopentameric protein where half the receptor is from a pH-gated bacterial channel, and the overall channel studied is activated by pH. We also do not aim to disparage this work, which we generally view as being very high quality and important for the field. Here, we briefly walk through the study's presented mutagenesis results. In Fig. 6, three mutants were studied in the PAM site, to test their effect on PS inhibition. At pH 5.5 and 10 μ M PS, the "WT" receptor is inhibited 60% by PS. Mutation of an essential PAM-site Q241 to Leu results in ~54% block by PS. W245L (the stacking Trp we discuss) results in loss of apparent NAM inhibition such that PS only blocks 40%. T305W, which was important for PAM potentiation (in the same study), gave a similar small loss of PS block as Q241L. Together these results suggest that mutating the PAM site may have a small but real either direct or indirect effect on pH activation and/or inhibition by PS. The study next shows mutations at the putative PS site. Here they look at two mutant constructs. First K390A. They note in the text that the side chain density for K390 is absent; we suggest that absent density means this side chain is conformationally disordered and unlikely to be making an important stabilizing interaction. Mutation of K390 to A results in no apparent change in PS IC₅₀ but does result in a decrease in maximal block amplitude, which is curious and the authors explore in modeling. In this panel of mutant experiments, the pH is lower (4.5); maximal block of the parent crystallization construct is 80% at max [PS]. For K390A, max block is 60%. The second and final proposed-NAM-site mutant construct studied contains 3 mutations in the proposed site: I391C, A398C, and F399C. This is a curious construct to study; cysteine is very hydrophobic and neither very long nor very short, and thus is not a major change from any of the residues it is replacing. Functionally, this mutation results in the same unchanged IC₅₀ (compared to the x-ray construct) and same 40% block as the K390I mutation. The authors summarize the results as showing no effect on PS block upon mutation of the PAM site, and significant change in PS block upon mutation of their proposed site. We disagree with the former and agree with the latter. We suggest that the modest effects from all of the mutants on PS inhibition suggest that neither of these sites is the key site for inhibition. It is also important to note that electrophysiology experiments by others, which we cite and discuss, demonstrate that sulfated neurosteroids cannot bind and inhibit when applied intracellularly, suggesting it would be

impossible for PS to access this proposed intracellular leaflet site in a biological setting.

We also encourage readers to look at the x-ray crystallographic maps. The density attributed to PS is very weak and noisy, while the unmodeled difference density at the PAM site is at least as strong if not stronger. Lastly, there is unmodeled density in the lower half of the pore that could result from occupancy by PS (though it is not very strong). Above, on the right, is one example interface, which has the strongest density in the PAM site (unmodeled) and comparatively weak density for the modeled PS.

In this context, I was wondering whether detection of rectification in the I/V curve in the presence of PS or DHEAS would be a way to provide stronger evidence of pore blocking as established in glutamate-gated NMDA receptors (10.1016/0896-6273(95)90049-7) or the inward-rectifier K⁺ channels (10.1146/annurev.physiol.66.032102.150822); since sulphate neurosteroids are charged, I would expect much more efficient inhibition by pore block in one voltage direction relative to the other.

We agree that observation of rectification would indeed be a strong argument to support a pore block mechanism. Others have done this already, and we repeated this kind of experiment, with the preliminary results shown below. They and we detect either little or no voltage-dependence of pore block. This result is a disappointment and may hint that we are still missing something. Or, as we suggest above, this ligand's affinity for the pore is due much more to other factors (negative charge likes to sit in an anion channel, hydrophobic steroid group likes to interact with the hydrophobic parts of the pore). It may be significant that the negatively charged sulfate group of both PS and DHEAS is oriented towards the extracellular side, likely the last moiety in either modulator to enter the pore, and potentially less determinant of permeation or binding than of solubility.

Minor points:

- Introduction: The Introduction is very short and much introductory information is given in the Results section. The authors might want to consider changing this organization, prepare the ground since the beginning, and focus only on the experimental evidence later.

We understand and explored different ways of writing up this study but settled on it being more interesting when bits of background are added along with some of the results. If there is a consensus that this organization is troubling, we can revise.

- Page 3: the sentence “to contrast GABA receptor binding...” reads odd.

We revised for clarity.

- Page 4: the authors refer to an a1b2y2 GABA receptor construct. What was modified in the construct relative to the physiological receptor? This important information should be given in the Main Text and not only in the Methods.

Great point, we have revised for clarity.

- Page 5: “hydrophobic stacking interactions”. Not sure it makes sense to talk about hydrophobic interactions in a non-polar environment like a lipid bilayer...

Another great point, revised.

- Page 5: the functional work showing the role of W246 could be described more explicitly in the Main Text (which mutations were done?). The same for L301 for which no Ref is given.

Certainly, we have made the text more specific:

“The indole side chain of W246 is positioned to make stacking interactions with the C and D rings of allopregnanolone, stabilizing the steroid in the binding pocket. The importance of this interaction is underscored by its mutation to leucine resulting in complete loss of potentiation by allopregnanolone and pregnanolone (19,32,33) with no effect on barbiturate potentiation (29).”

We do not know of L301 mutagenesis results.

- Figures: Fig.2-3-4-S5-S6 are rather dark and way too small so details (including residue numbering and interactions) are difficult to appreciate.

We are going to check with the editor about their preferences in addressing this question.

- Page 7: the sentence starting with “We conclude...” comes too early and should be moved later in the story.

We changed the word “conclude” to “propose.”

- Page 7: why aW245 and not aW246 is discussed here as done in Fig.2?

Thank you for catching this. We have corrected W245 to W246.

- Page 7: “presence of sticky steroid binding site”. Since electron-density was observed in cryo-EM, which steroid could be possibly present during sample preparation?

In the cryo-EM sample, there is no bona fide lipid bilayer. As such, this density may arise from bound sulfated steroid wherein the sulfate is oriented into solvent or toward lipid head groups.

- Page 7: “Supp Fig5e-h” should be “Supp Fig5i-j”

On p. 7, we changed the call for Supp Fig 5f-g to Supp Fig. 5e-h. We are guessing this is what you were suggesting, as our original call there was incorrect.

- Page 7: “neither mutations in the a-y interface... affected inhibition by PS”. How significant are the differences between H102R and WT observed in Fig.S6?

We see what you are looking at: in this pair of example traces, it looks like PS is slightly less efficacious at blocking peak and steady state currents in H102R compared to the parent EM construct. To address this potential discrepancy, we analyzed the data from all the recordings, and show the results here (below). There is no significant difference between the WT and mutant responses to PS. We now include a formatted version of this analysis as a final new panel in Supplementary Figure 6.

Comparison of PS inhibition of the peak and steady state currents of EM construct and mutants. The bars indicate mean ± SD, n = 5(EM,) 5 (H102R) and 6 (H102A) with individual cells.

- Page 8: the introduction paragraph on how lipids help fixing neurosteroids to GABAR as part of their binding site is misleading since lipid densities are discussed later in the subsection in competition with neurosteroid binding.

We are not sure, but we discuss how the lipids apparently occupy a site of PAMs such as etomidate and propofol in the neurosteroid structures. In our observation we do not see lipids competing with neurosteroid binding. We have added language clarifying this point in the text, and thank the reviewer for highlighting the need for revision to make our point better understood.

- Page 9: "GABA:ELIC chimera" should be "GABA:GLIC chimera" as discussed on page 12

Thank you; corrected.

- Page 11: Add Ref to the work of Lavery et al (10.1038/nsmb.3477) on the first sentence.

Done.

- Page 12: CHS should be spelled in full

Done.

Reviewers' Comments:

Reviewer #1:

Remarks to the Author:

The authors have addressed my concerns adequately.

Reviewer #2:

Remarks to the Author:

The revised manuscript by Legesse et al addresses/clarifies most of my critical points. Some of them, however, were not fully addressed and the mechanistic conclusions, albeit intriguing, remain speculative.

Concerning the mechanism of potentiation, the new potential of mean force calculations in the Fig 3j convincingly show that the pore configuration stabilized by ALLO binding favors chloride permeation at 9' relative to GABA-only, which supports the proposed pore dilation mechanism. However, this mechanistic conclusion is drawn from the analysis of structures that are supposed to be non-conducting and remains, in my opinion, speculative. Therefore, I would like the authors to clearly state in the Text (i.e., Results and Discussion) that conclusions on pore dilation were drawn from structures representative of non-conductive, desensitized states and should be taken with care.

Concerning the mechanism of inhibition, although I appreciated the new experiments to test for rectification as well as the extensive analysis of the structural and mutagenesis data of Laverty et al (10.1038/nsm.3477), which helps the reader understanding the possible causes of the controversy, I am still not convinced by the absence of voltage-dependent inhibition in a pore-blocking mechanism. Although I understand the authors' argument to explain this observation (i.e., a tradeoff between hydrophobic interactions in the pore and voltage-dependent interactions with the charged sulfate), no evidence for it is provided and this conclusion remains speculative. By quoting the authors in the reply, the absence of rectification in presence of PS "is a disappointment and may hint that we are still missing something". If so, I would tone down the narrative in the Discussion stating that the question is not settled yet, and further studies are required. Following the authors rationale, perhaps functional studies of mutants with polar or less hydrophobic residues in the pore (e.g., V257S or A252S), which are expected to lower the binding affinity for PS in the pore, will show stronger and detectable voltage-dependent inhibition, thus demonstrating pore-blocking by sulfated neurosteroids.

Last, since the structure stabilized by PS and DHEAS is presented as a non-conductive state of the GABA receptor (page 9), I am wondering whether the mechanism of inhibition emerging from these structures is consistent with pore blocking or rather it could/should be interpreted as a conformational selection of the desensitized state by neurosteroid binding into the pore. In this case, sulfated neurosteroids would be described as "desensitizers" rather than "pore blockers", one more reason for caution on drawing mechanistic conclusions on allosteric modulation.

Finally, I find it is a pity that changes in the revised Text were not highlighted, nor explicitly reported in the reply letter, which made my search a bit more difficult. Also, judging conclusions without being able to look at coordinates was difficult for me and high-resolution structures should have been at least confidentially released for the reviewers. Unfortunately, I did not have this opportunity.

We thank both peer reviewers for their constructive comments, suggestions, and questions in the first round of review and Rev. 2 for the additional comments, which we respond to here. Below we provide point-by-point responses in blue font.

Reviewer #1

The authors have addressed my concerns adequately.

Reviewer #2

The revised manuscript by Legesse et al addresses/clarifies most of my critical points. Some of them, however, were not fully addressed and the mechanistic conclusions, albeit intriguing, remain speculative.

Concerning the mechanism of potentiation, the new potential of mean force calculations in the Fig 3j convincingly show that the pore configuration stabilized by ALLO binding favors chloride permeation at 9' relative to GABA-only, which supports the proposed pore dilation mechanism. However, this mechanistic conclusion is drawn from the analysis of structures that are supposed to be non-conducting and remains, in my opinion, speculative. Therefore, I would like the authors to clearly state in the Text (i.e., Results and Discussion) that conclusions on pore dilation were drawn from structures representative of non-conductive, desensitized states and should be taken with care.

We agree that activated state structures would also be helpful, and without them the structural mechanism is incomplete. However, the findings on pore widening and subsequent hydration, and consistency of this result among many structures with PAMs bound, suggests it is a robust/meaningful component of potentiation. We added this statement to the results: "A caveat of these interpretations is that all PAM complexes studied to date adopt non-conducting, desensitized-like conformations." We added this statement to the discussion: "An activated state structure would shed further light on PAM mechanism."

Concerning the mechanism of inhibition, although I appreciated the new experiments to test for rectification as well as the extensive analysis of the structural and mutagenesis data of Laverty et al (10.1038/nsmb.3477), which helps the reader understanding the possible causes of the controversy, I am still not convinced by the absence of voltage-dependent inhibition in a pore-blocking mechanism. Although I understand the authors' argument to explain this observation (i.e., a tradeoff between hydrophobic interactions in the pore and voltage-dependent interactions with the charged sulfate), no evidence for it is provided and this conclusion remains speculative. By quoting the authors in the reply, the absence of rectification in presence of PS "is a disappointment and may hint that we are still missing something". If so, I would tone down the narrative in the Discussion stating that the question is not settled yet, and further studies are required. Following the authors rationale, perhaps functional studies of mutants with polar or less hydrophobic residues in the pore (e.g., V257S or A252S), which are expected to lower the binding affinity for PS in the pore, will show stronger and detectable voltage-dependent inhibition, thus demonstrating pore-blocking by sulfated neurosteroids.

We agree; thank you. We added the following two sentences to the discussion. "Still, the absence of voltage-dependence of block remains a weak point in our proposed pore block mechanism. Mutagenesis to render the pore more polar in regions we propose the steroid interacts could further test this mechanism."

Last, since the structure stabilized by PS and DHEAS is presented as a non-conductive state of the GABA receptor (page 9), I am wondering whether the mechanism of inhibition emerging from these structures is consistent

with pore blocking or rather it could/should be interpreted as a conformational selection of the desensitized state by neurosteroid binding into the pore. In this case, sulfated neurosteroids would be described as “desensitizers” rather than “pore blockers”, one more reason for caution on drawing mechanistic conclusions on allosteric modulation.

We think it is reasonable to call any antagonist that exerts its activity by occluding the ion permeation pathway a pore blocker. For example, picrotoxin is universally agreed to be a pore blocker, yet is known to stabilize both resting and activated states, depending on the absence/presence of GABA, thereby aiding in recovery from desensitization. We propose that a ligand that blocks the pore can be called a pore blocker regardless of the conformational state(s) it stabilizes. We already included a statement in the results making it clear that the conformational state observed in the presence of sulfated neurosteroids is not identical to the desensitized state observed when only GABA is present. “Interestingly, we found that binding of sulfated neurosteroids in the pore stabilized a channel conformation distinct from that stabilized by GABA alone (Supplementary Fig. 7), consistent with a recent functional study concluding that PS and DHEAS stabilize a novel nonconducting state³⁸.”

Finally, I find it is a pity that changes in the revised Text were not highlighted, nor explicitly reported in the reply letter, which made my search a bit more difficult. Also, judging conclusions without being able to look at coordinates was difficult for me and high-resolution structures should have been at least confidentially released for the reviewers. Unfortunately, I did not have this opportunity.

This would have frustrated us as reviewers also. Please understand that we did submit a version with all changes tracked and displayed. We assumed this would have been available to you- that was our intention. We should have done better in pasting the exact changes into the reviewer response letter.

When we learned that the manuscript would be sent for peer review, we immediately shared a folder containing all models and maps with the editor. We shared this again the day we saw these comments. Please know that we did make these documents available to the journal (early on, and again later), and they should have made their way to you, especially upon request.